

# *D* region ion-neutral coupled chemistry within a whole atmosphere chemistry-climate model

Tamás Kovács[1], John M. C. Plane[1], Wuhu Feng[1,2], Tibor Nagy[3], Martyn P. Chipperfield[2], Pekka T. Verronen[4], Monika Andersson[4], David A. Newnham[5], Mark A. Clilverd[5] and Daniel R. Marsh[6]

[1] School of Chemistry, University of Leeds, Leeds, LS2 9JT, UK
[2] School of Earth and Environment, University of Leeds, Leeds, LS2 9JT, UK
[3] IMEC, Research Centre for Natural Sciences, Hungarian Academy of Sciences, Budapest, Hungary
[4] Finnish Meteorological Institute, Helsinki, Finland
[5] British Antarctic Survey, Cambridge, CB3 0ET, UK
[6] National Centre for Atmospheric Research (NCAR), Boulder, Colorado, USA

*Correspondence to*: Tamás Kovács (takovacs@gmail.com)

**Abstract.** This study presents a new ion-neutral chemical model coupled into the Whole Atmosphere Community Climate Model (WACCM). The ionospheric *D* region (altitudes ~50 - 90 km) chemistry is based on the Sodankylä Ion and Neutral Chemistry (SIC) model, a 1-dimensional model containing 306 ion-neutral and ion-recombination reactions of neutral species, positive and negative ions, and electrons. The SIC mechanism was reduced using the Simulation Error Minimization Connectivity Method (SEM-CM) to produce a reaction scheme of 181 ion-molecule reactions. This scheme describes the concentration profiles at altitudes between 20 km and 120 km of a set of major neutral species ($HNO_3$, $O_3$, $H_2O_2$, $NO$, $NO_2$, $HO_2$, $OH$, $N_2O_5$) and ions ($O_2^+$, $O_4^+$, $NO^+$, $NO^+(H_2O)$, $O_2^+(H_2O)$, $H^+(H_2O)$, $H^+(H_2O)_2$, $H^+(H_2O)_3$, $H^+(H_2O)_4$, $O_3^-$, $NO_2^-$, $O^-$, $O_2$, $OH^-$, $O_2^-(H_2O)$, $O_2^-(H_2O)_2$, $O_4^-$, $CO_3^-$, $CO_3^-(H_2O)$, $CO_4^-$, $HCO_3^-$, $NO_2^-$, $NO_3^-$, $NO_3^-(H_2O)$, $NO_3^-(H_2O)_2$, $NO_3^-(HNO_3)$, $NO_3^-$ $(HNO_3)_2$, $Cl^-$, $ClO^-$), which agree with the full SIC mechanism within a 5% tolerance. Four 3D model simulations were then performed, using the impact of the January 2005 Solar Proton Event (SPE) on *D* region $HO_x$ and $NO_x$ chemistry as a test case of four different model versions: the standard WACCM (no negative ions and a very limited set of positive ions); WACCM-SIC (standard WACCM with the full SIC chemistry of positive and negative ions); WACCM-D (standard WACCM with a heuristic reduction of the SIC chemistry, recently used to examine $HNO_3$ formation following an SPE); and WACCM-rSIC (standard WACCM with a reduction of SIC chemistry using the SEM-CM Method). Standard WACCM misses the $HNO_3$ enhancement during the SPE, while the full and reduced model versions predict significant $NO_x$, $HO_x$ and $HNO_3$ enhancements in the mesosphere during solar proton events. The SEM-CM reduction also identifies the important ion-molecule reactions that affect the partitioning of odd nitrogen ($NO_x$), odd hydrogen ($HO_x$), and $O_3$ in the stratosphere and mesosphere.



## 1 Introduction

Energetic charged particles that impact on the Earth's atmosphere come from several different sources: in the case of protons (and some heavier ions), directly from the Sun during Solar Proton Events (SPEs) and from outside the Solar System in the form of high-energy Galactic cosmic rays; and in the case of electrons, from the radiation belts around the Earth during geomagnetic storms and sub-storms (Du et al., 2008; Kurt et al., 2004; Shea and Smart, 1992; Sinnhuber, 2012). Energetic particle precipitation (EPP) can lead to significant perturbations of the chemical composition from the lower thermosphere all the way down to the stratosphere (Jackman et al., 2014; Verronen et al., 2011). Protons with energy above 1 MeV can penetrate down to the mesosphere and the upper stratosphere, particularly at high geomagnetic latitudes. Energetic Particle Precipitation (EPP) causes ion-pair formation, and the subsequent neutralization produces odd nitrogen ($NO_x$= N + NO + $NO_2$) and odd hydrogen ($HO_x$= H + OH + $HO_2$) species. The $NO_x$ and $HO_x$ species destroy mesospheric ozone via catalytic cycles (Crutzen, 1970; McElroy et al., 1992), which can have a significant impact on the radiative balance of the middle atmosphere and hence on climate (Sinnhuber, 2012).

Current whole atmosphere chemistry climate models such as the Hamburg Model of the Neutral and Ionized Atmosphere (HAMMONIA) (Schmidt et al., 2006) and the Whole Atmosphere Community Climate Model (WACCM) (Garcia et al., 2007; Jackman et al., 2011; Marsh et al., 2007; Marsh et al., 2013) have simple lower $E$ region plasma chemistry, essentially $NO^+$ and $O_2^+$ ions balanced by free electrons, but do not describe the much greater complexity of the $D$ region where clusters and negative ions dominate (Brasseur and Solomon, 2005; Sinnhuber, 2012; Winkler et al., 2008). Therefore, in order to model the impacts of EPP in the atmosphere below 90 km, it is essential to have a detailed treatment of $D$ region ion chemistry.

The leading kinetic model of $D$ region chemistry is the Sodankylä Ion and Neutral Chemistry (SIC) model, developed jointly at the Sodankylä Geophysical Observatory and the Finnish Meteorological Institute (Turunen et al., 1996; Verronen et al., 2011). It contains 306 ion-molecule and also 2254 ion-ion recombination reactions. The model is a 1D model with simple vertical transport (molecular and eddy diffusion) to balance the computational cost. Putting such a large number of additional reactions into a 3D global model could be computationally expensive because of the large number of stiff partial differential equations that need to be solved. In view of this, several of the authors of the present paper carried out a heuristic reduction of the full SIC chemistry scheme - i.e., based on chemical knowledge and intuition - in order to produce a $D$ region ion scheme of 196 ion-neutral reactions, which was able to account for the substantial enhancements of $HNO_3$ that have been observed between ~50 and 80 km after major SPEs (Andersson et al., 2016).

Encouraged by the development of WACCM-D, we have now carried out a systematic mechanism reduction of the SIC chemistry, which is the subject of this paper. The objective of the reduction was to model a set of important species during a highly perturbed period - the intense SPE of late October 2003 - at an acceptable level of accuracy, but with reduced computational time. The reason for choosing a highly perturbed period is that with this it was possible to reveal every important chemical detail, while using a weaker event would have caused some important reactions to be missed. The full





and reduced SIC model chemistries were then tested in WACCM, where the medium SPE of 15-17 January 2005 was used as a test case of how well the reduced scheme captures the substantial atmospheric perturbation to $NO_x$, $HO_x$ and $HNO_3$.

## 2 Methodology

The SIC model (Verronen et al., 2011) contains 306 ion-neutral reactions in total. These are listed in the Supplementary Material (SM), where the reactions in the grey shaded rows are the subset selected systematically in this study for the reduced model, and the reactions shown in bold type are in WACCM-D (Andersson et al., 2016). It should be noted that rate coefficients for some of the ion-molecule reactions have been updated using a recent review (Pavlov, 2014) - these changes are documented in the SM.

### 2.1 Description of the 1D SIC model

In the initial step, primary proton collisions with molecular nitrogen cause dissociative ionization and produce secondary electrons which also form atomic nitrogen from molecular nitrogen. A detailed description of the SIC model is given in (Verronen et al., 2005), where the ionization scheme is also described. It should be noted that the SIC model assumes that the electron temperature is the same as that of the neutral atmosphere, which is a reasonable simplification in the *D* region (Roble, 1976). The model solves time-dependent concentrations between 20 km and 150 km with 1 km vertical resolution. The concentrations are controlled by solar radiation, particle precipitation, chemical reactions and vertical transport. Daily solar irradiance data are taken from the SOLAR2000 model (Tobiska and Bouwer, 2006). The integrated proton fluxes measured by the Geostationary Operational Environmental Satellite (GOES-11) satellite are converted to energy resolved flux spectra using the exponential rigidity relation (Freier and Webber, 1963). Ionization rates are calculated from the spectra based on the proton energy-range measurements in standard air (Bethe and Ashkin, 1953) as described in (Verronen et al., 2005), assuming that 35 eV of energy is required to produce one ion pair (Porter et al., 1976). The SIC model inputs ionization rates as a function of time and pressure, choosing 3-hour time resolution. The MSISE-90 model (Picone et al., 2002) is used for climatological average altitude profiles of $N_2$, $O_2$ and temperature for the background atmosphere, and mid-latitude concentrations of $N_2O$, $H_2$, $HNO_2$, $HCl$, $Cl$, $ClO$, $CH_3$, $CH_4$, $CH_2O$ and $CO$ at altitudes of 10, 15, 20, 25, 30, 45, 60 and 100 km are taken from (Shimazaki, 1984).

### 2.2 Mechanism reduction

The updated SIC mechanism was reduced using the Simulation Error Minimization Connectivity Method (SEM-CM) (Nagy and Turanyi, 2009). SEM-CM is based on the Connectivity method (Turanyi, 1990; Turanyi et al., 1989), which identifies *necessary* species based on their kinetic connectivity to the designated *important* species. The important species are those whose concentration should be accurately reproduced by the reduced model. Further necessary species are those whose inclusion in the reduced mechanism is also needed to achieve this aim.





As a result, the reduced mechanism contains only the reactions of the important and the necessary species – that is, the non-redundant reactions. The set of redundant species take part only in the redundant subset of reactions. In addition, in the mechanism reduction process the redundant species are identified. In the following, redundant reactions refer to the set of reactions involving redundant species.

Kinetic connectivity is defined via the kinetic differential equation which describes the change of species concentrations (vector $\mathbf{f} = (f_1,...,f_N)$ for $N$ species) due to thermal ($\tau$ number of reactions) and photochemical ($\pi$ number of reactions) transformations:

$$\frac{d\mathbf{c}}{dt} = \mathbf{f}\big(\mathbf{c}, \mathbf{k}(T, p), \mathbf{J}(F(\lambda), T, p)\big) \tag{E1}$$

Vector function $\mathbf{f}$ defines the rate of concentration change of all species as a function of the concentration of all species
($\mathbf{c}=(c_1,...,c_N)$) and the thermal and photochemical rate parameters (vectors $\mathbf{k}=(k_1,...,k_\tau)$ and $\mathbf{J}=(J_1,...,J_\pi)$ respectively), which depend on temperature ($T$), pressure ($p$) and solar actinic flux (function $F(\lambda)$, where $\lambda$ is wavelength). The dimension of $\mathbf{J}$ is 23, which equals the number of photodissociation (16) and photoionization (7) reactions, while the dimension of $\mathbf{k}$ is the total number of neutral-neutral and ion-neutral thermal reactions (306). The kinetic connectivity $B_j$ of species $j$ to the set of $i$ selected species is measured by the sum of the squared elements of the log-normalized Jacobian matrix ($\bar{J}_{ij} = \partial \log f_i / \partial \log c_j$,
where matrix $\bar{\mathbf{J}}$ is an $N \times N$ matrix) of the kinetic differential equation.

$$B_j = \sum_{i \, \text{selected}} \bar{J}_{ij}^2 = \sum_{i \, \text{selected}} \left( \frac{\partial \log f_i}{\partial \log c_j} \right)^2 \tag{E2}$$

Species with relatively large $B_j$ values are closely linked to the set of selected species (which initially are the important species), and their presence is necessary in the mechanism for the formation and consumption of the selected species. The Jacobian matrix is determined at several time points from a simulation with the full model, and then stored for the iteration.
The reduced mechanism is constructed in an iterative procedure by adding additional species to the set of selected species in order to achieve the stipulated accuracy with which the concentrations of the important species need to be reproduced. Note that the initial set of necessary species is the set of important species for which the concentrations are to be reproduced within the given accuracy.

However, adding a single species to the important ones will not necessarily require the inclusion of additional reactions, thus
species should be added in so-called complementary sets. A complementary set contains all species from a reaction which has not yet been selected. The kinetic connectivity of the $k$-th complementary set ($C_k$) with $n_k$ species to the selected species is defined as the average of the $B$ values:

$$C_k = \frac{1}{n_k} \sum_{j \in \text{set } k} B_j \tag{E3}$$





At each time of the investigation, the complementary sets are ranked according to their connectivity and several reduced mechanisms are formed by adding the strongly connected sets to the selected species and including their reactions in the reduced scheme. The SEM-CM mechanism at this step investigates whether each species in the reduced mechanism is either an initially present species or there is a chemical pathway of formation from the species initially present. Additional

complementary sets are added until this condition is fulfilled (Nagy and Turanyi, 2009). The mechanism reduction is carried out in a gradual species inclusion procedure ("building"), and in each step of this, several candidate extended sets of species and corresponding reduced mechanisms are generated. The candidate reduced mechanisms are simulated and stored with their simulation errors in a database sorted by their numbers of species. In the database the mechanisms with one more species and the one with smallest simulation error is selected. If the mechanism fulfils the required accuracy criterion, the

reduction procedure is complete. If this is not the case, the SEM-CM building step is repeated with these "selected" species of smallest error until the required accuracy is met. In essence, the construction of the reduced mechanism according to the SEM-CM procedure is governed by the steepest decrease of simulation error.

The aim of the mechanism reduction procedure is to reduce the maximum error to below a defined threshold. To achieve this, simultaneous minimization of a maximum ($\delta_{\text{MAX}}$) and a root-mean-square global error ($\delta_{\text{RMS}}$) by two SEM-CM threads

was found to be very efficient. These errors were defined as:

$$\delta_{\text{MAX}} = \max_{i:\text{scenarios}} \; \max_{j:\text{species}} \; \max_{k:\text{time points}} \left| \delta_{ijk} \right| \tag{E4}$$

$$\delta_{\text{RMS}} = \operatorname*{rms}_{i:\text{scenarios}} \; \operatorname*{rms}_{j:\text{species}} \; \operatorname*{rms}_{k:\text{time points}} \left| \delta_{ijk} \right| \tag{E5}$$

Here $\delta_{ijk}$ is a local dimensionless error measuring the deviation between the concentrations of important species $j$ at time point $k$ in scenario $i$ modelled by the full ($c_{ij}^{\text{full}}(t_k)$) and the reduced mechanisms ($c_{ij}^{\text{red}}(t_k)$). A mixed type of local error

defined by (Nagy and Turanyi, 2009) was used:

$$\delta_{ijk} = 2 \frac{c_{ij}^{\text{red}}(t_k) - c_{ij}^{\text{full}}(t_k)}{c_{ij}^{\text{full}}(t_k) + c_{ij,\text{MAX}}^{\text{full}}} \approx \begin{cases} \dfrac{c_{ij}^{\text{red}}(t_k) - c_{ij}^{\text{full}}(t_k)}{c_{ij}^{\text{full}}(t_k)} & \text{if } c_{ij}^{\text{full}}(t_k) \sim c_{ij,\text{MAX}}^{\text{full}} \\[3mm] \dfrac{c_{ij}^{\text{red}}(t_k) - c_{ij}^{\text{full}}(t_k)}{c_{ij,\text{MAX}}^{\text{full}} / 2} & \text{if } c_{ij}^{\text{full}}(t_k) << c_{ij,\text{MAX}}^{\text{full}} \end{cases} \tag{E6}$$

This mixed error behaves as a relative error close to the maximum concentration of species $j$ in scenario $i$ ($c_{ij,\text{MAX}}^{\text{full}}$) obtained with the full mechanism, whereas at much lower concentrations it damps large relative deviations and behaves as a scaled absolute error.

For the SIC mechanism reduction, the maximum of a large SPE was selected in order to be able to ensure that the reduced scheme could satisfactorily model large, short-lived perturbations to ions and neutrals. This was the SPE in October 2003, the "Halloween Storm" (Burlaga et al., 2005) that peaked at 0615 UT on 29[th] October 2003 with a proton flux of 29,500



(pfu>10MeV) (pfu: particle flux unit = particle cm$^{-2}$ ster$^{-1}$ s$^{-1}$). Four specific altitudes, 60, 70, 80 and 90 km were selected to represent the *D* region. The selection of important chemical species contained those that are the major neutral and ionic species according to a standard 1D SIC run. The aim of the reduction was to find the smallest sub-mechanism which has a maximum root-mean-square error ($\delta_{RMS}$) of important species at each of the four altitudes less than 5% (i.e. $\delta_{RMS}<0.05$). Note

that the 5% accuracy is guaranteed only for the selected 1D conditions (altitudes = 60, 70, 80, 90 km) but not for any other conditions. The 1D model was run for 10 days and local errors were calculated at 100 points distributed logarithmically between 1 s and 10 days (864,000 s), with an increase of approximately 15% between adjacent times. In order to set the proton flux to peak, the reduction was done at the peak time of the storm (0615 UT, 29$^{th}$ October 2003).

**2.3 3D modelling using WACCM**

WACCM is a comprehensive numerical model extending vertically from the ground up to the lower thermosphere (~140 km) and is part of the NCAR Community Earth System Model (CESM) (Hurrell et al., 2013). Here we use the specified dynamics (SD) version of WACCM 4 (Marsh et al., 2013), which has 88 pressure levels from the surface to 5.96×10$^{-6}$hPa and a horizontal resolution of 1.9$^{o}$ × 2.5$^{o}$ (latitude × longitude). The model contains all the important processes in the

15 atmosphere: chemistry, radiative transfer, auroral processes, non-local thermodynamic equilibrium, ion drag, SPE ionization rate from 1963 to 2016 and the molecular diffusion of the constituents. In the SD version the model is forced with ECMWF meteorology re-analysis (Dee et al., 2011) from surface to 50 km. 1% of the meteorological conditions (temperature, winds, surface pressure, specific humidity, surface wind stress, latent, sensible heat flux) were combined with the WACCM fields below 50 km at every model dynamics time step. This nudging factor then reduces linearly from 1% to 0% between 50 km

and 60 km. Above 60 km there is no nudging to the re-analysis fields and the model is free-running. The EPP caused NO$_x$ production was determined from the ionization rate ($I$) that is proportional to the energy deposition rate, and the NO$_x$ production rate is expressed as 1.25×$I$. This is the standard parameterization in WACCM. For HO$_x$ there was no similar parameterization involved.

Both the full SIC model and its reduced version, rSIC, were included in WACCM to produce the WACCM-SIC and

25 WACCM-rSIC models, respectively. WACCM-rSIC included the WACCM neutral reactions and the rSIC module which has a similar number of reactions (181) as WACCM-D (196) but they differ with the reactions of O$_2^+$ clusters and larger proton hydrates (see the SM). As ion-ion reactions were not included in the reduction process, they were considered in a different way for WACCM-D, WACCM-SIC and WACCM-rSIC. WACCM-D has 112 two-body and 14 three-body reactions. The two-body reactions involves the reactions of O$_2^+$, H$^+$(H$_2$O)$_n$ ($n$ = 3, 4, 5) and NO$^+$(H$_2$O)$_m$ ($m$ = 0, 1, 2) with a range of

30 negative ions (CO$_3^-$, HCO$_3^-$, CO$_4^-$, O$_2^-$, NO$_3^-$, NO$_2^-$, Cl, NO$_3^-$(H$_2$O)$_n$ ($n$ = 1, 2), Cl-(H$_2$O), NO$_3^-$(HCl), NO$_2^-$(H$_2$O), CO$_3^-$(H$_2$O), Cl$^-$(HCl) and NO$_3^-$(HNO$_3$)). In the case of WACCM-rSIC the three major positive and negative ions (NO$^+$, H$^+$(H$_2$O)$_3$, H$^+$(H$_2$O)$_4$, O$_2^-$, CO$_3^-$ and NO$_3^-$) were selected (based on the results from the 1D model run for January 2005), and only the two-body ion-ion reactions of these species with the non-redundant ion species (of opposite charge) were included. For WACCM-SIC, the two-body ion-ion reactions of these three major positive and three major negative ions with all ions of





opposite charge were included. This is a good approximation as this still allows the ion-ion channels to contribute to the net ion balance but ignores those channels where the contributing ion concentrations do not dominate. Three-body ion-ion recombination reactions are many orders of magnitude slower than the two-body processes in the $D$ region, and hence were not included in both the full and the reduced 3D models.

After a 5-year spin up, all four models – WACCM, WACCM-SIC, WACCM-rSIC and WACCM-D - were run for 30 days from 1st January 2005, a period encompassing a significant SPE which reached a maximum on 17th January 2005 at 17:50 UT. The January 2005 event (15th January 2005 – 20th January 2005) was chosen in order to test the accuracy of the reduced model for a different scenario to the Halloween storm. The reason for choosing this event was to be consistent with our earlier study using WACCM-D (Verronen et al., 2015).

**3 Results and discussion**

The concentration tolerance for the important species was set to 5%, meaning that the concentrations of the important species in the reduced model are reproduced to within 5% of the full model. The change of root-mean-square (rms) error as a function of number of species is shown in Figure 1, where it is clear that the error drops initially rapidly with increasing number of species and then continues to decrease more slowly. The species were added in order of decreasing error. Until

the reaction pathways are built up completely in the reduced mechanism, various feedback effects are not necessarily manifested in the concentration profiles of the important species. The intermediate reduced mechanism will therefore have one or more inactive truncated pathways. A sudden decrease in the rms error is observed after a long stagnation when those species are added which close the pathways; their early inclusion would not give a smaller reduced mechanism, as then other species would be missing from the pathway. The jump in the numbers of necessary species from 60 km to 70 km is

connected to the increased concentrations of $NO^+$ and $O_2^+$- these ions are involved in several ion-molecule reactions. The sets of important and necessary species are tabulated in Table 1. These were selected on the basis of which species are expected to be chemically directly associated with NO, $NO_2$, $HNO_3$ and $HO_2$; the five most abundant positive and negative ions were essential to be included. Ions were also selected in the initial set of important species. When a neutral-only list of important species was tried the reduction never reached the required accuracy which indicated that ions are crucial part for

the balance of the major neutrals via neutral-ion interconnection. Bold-face-font species indicate the initial set of important species, while the others are the necessary species selected through the reduction process. As electrons are not important charge carriers in the $D$ region, they were not considered as important species. The reduced 1D model decreases the computational time by 25% compared to the full model, while the number of reactions dropped from 306 to 181 and the number of species from 129 to 81. However, more importantly, the results of the reduction process makes it possible to

identify the reactions of the important and necessary species that affect the atmospheric $NO_x$ and $HO_x$ concentrations, and can therefore impact on stratospheric ozone levels.



### 3.1 1D modelling

Figure 2 compares the major neutral and ion profiles produced by the SIC and rSIC models for the SPE conditions of January 2005. As expected, the concentrations of important neutral and ionic species are reproduced very well by the reduced model, within the height range (60 - 90 km) where the reduction was carried out. Figure 2 also shows the relative

errors of the mechanism reduction on the concentration profiles. It is clear that the errors remain within the 5% tolerance, except for $HO_2$ below 30 km and for $O_2^-$ above 115 km, but these are outside the *D* region. Furthermore, the neutral profiles are reproduced within a factor of 2 under *any* conditions, even above 80 km. This is not surprising as the chemical scheme in the lower thermosphere is much simpler, being governed by $NO^+$, $O_2^+$, $O^+$ and electrons (Danilov, 1994).

All photodissociation and photoionization reactions are found to be important, and are therefore included in the 181 reactions

in the reduced model. 27% of the positive ion recombination reactions could be eliminated, which are those involving recombination of $NO^+(N_2)$, $NO^+(CO_2)$ and their mono- and di-hydrated complex clusters. The 7 and 8 $H_2O$-containing proton hydrates are also redundant species. All the other reactions involving electrons, including photodetachment and reactive electron detachment are important. This will be discussed later in more detail. The reduction was clearly most effective for the positive ions with 55 out of the 124 positive ion reactions (i.e. 44%) being redundant, including: the

reactions of $CH_3CN$ and all its clusters; all the proton hydrate + $N_2O_5$ reactions; and the reactions of proton-hydrate-$HNO_3$ complex clusters with $N_2O_5$ and/or $H_2O$. The majority of the redundant species are the hydrated clusters, and only a few $N_2$ and $CO_2$ clusters are included, such as the reactions of $NO^+(N_2)$ and $NO^+(CO_2)$.

For the negative ions, 54 out of the total of 124 reactions (i.e. 43%) were redundant, which is similar to the subset of redundant positive ion reactions. Similar to the positive ion results, the redundant list mostly arises from the reactions of

20 hydrate clusters. From the reduced set it can be concluded that in general the majority of the positive ion reactions are the reactions of $NO^+$ or $O_2^+$ and their hydrated clusters. The reactions of proton-hydrate clusters up to $n = 6$ also turn out to be necessary under all conditions, apart from the $H^+(H_2O)_n + N_2O_5$ reactions; the hepta- and octa-hydrates are both redundant species. The mechanism reduction in the vicinity of the boundary between the mesosphere and lower thermosphere is more efficient, with 60% of the reactions found to be redundant. This is not surprising as lower thermospheric chemistry is

25 governed chiefly by $NO^+$ and $O_2^+$, and the scheme simplifies as the negative ion concentrations (principally $O_2^-$, $CO_3^-$ and $NO_3^-$) decrease by many orders of magnitude from 80 km to 90 km. The reactions of $HCO_3^-$ with neutrals are redundant under all conditions. $OH^-$ generation and consumption are important only at ~60 km, and above this height $OH^-$ becomes redundant. The reactions of $CO_4^-$ are redundant already between 70 km and 80 km, while the $NO_2^-$ reactions are redundant between 80 km and 90 km. The total number of reactions in the reduced model is 181. Testing the reduced mechanism with

30 the 3D model is described below.



## 3.2 3D WACCM modelling

The full and reduced SIC models were coupled into WACCM to produce WACCM-SIC and WACCM-rSIC, respectively. The standard WACCM model contains only the five major positive ions ($N^+$, $N_2^+$, $O^+$, $O_2^+$ and $NO^+$) and has no negative ions. WACCM-D is described in our recent paper (Verronen et al., 2016). The WACCM-D reactions are shown in bold typeface in the SM. From this it is clear that the majority of reactions in WACCM-rSIC and WACCM-D are the same. However, WACCM-D omits the reactions of the larger proton hydrates and their clusters ($H^+(H_2O)_6$, $H^+(H_2O)_n(CO_2)$, $H^+(H_2O)_n(N_2)$ ($n = 1, 2$), $H_3O^+(OH)(CO_2)$ and $H_3O^+(OH)(H_2O)$)), and the reactions of $O_2^+$ clusters ($O_2^+(N_2)$, $O_2^+(CO_2)$, $O_2^+(H_2O)(N_2)$, $O_2^+(CO_2)(N_2)$ and $O_2^+(H_2O)_2$). These four WACCM-based models (WACCM, WACCM-D, WACCM-SIC and WACCM-rSIC) were tested for the 17[th] January 2005 SPE, a moderate event with peak flux of 5040 (pfu>10MeV) at 17:50 UTC. For auroral electrons and SPEs, the vertical profile of the energy deposition is inferred from electron and proton fluxes, observed in low earth orbit and in geostationary orbit, respectively (Holt, 2013).

We investigated the effect of the SPE on $O_3$ and the major $NO_x$ and $HO_x$ species. The results are shown in Figure 3-7, for northern hemisphere polar latitudes (60° - 90°N) during January 2005. Figure 3a shows the calculated polar $O_3$ mixing ratio from the four WACCM based models for January 2005; the percentage differences from WACCM-SIC are shown in Figure 3b. It is important to note that the 5% accuracy was guaranteed only for the selected 1D conditions. Therefore the deviation can be quite different at other conditions which means that the > 5% accuracy with the 3D model is not surprising. On the other hand, the different treatments of the ion-ion reactions in WACCM-SIC and WACCM-rSIC could also increase the deviations – *ie.* only the reactions of the major positive and negative ions with the non-redundant species were selected for the WACCM-rSIC, while for WACCM-SIC all the major positive and negative ion reactions were considered. Inspection of this figure shows generally very good agreement (average difference <10%, maximum difference ±30%) between WACCM-rSIC and WACCM-SIC. There are somewhat larger differences between WACCM-D and WACCM-SIC, with an overestimate of the $O_3$ concentration by up to 100% between 65 and 70 km. This is due to the lack of proton hydrate and $O_2^+$ cluster reactions in WACCM-D. Note that the standard WACCM fails to simulate the $O_3$ depletion between 55 and 70 km during the SPE (January 16[th] - 23[rd]), which illustrates the effect of not including positive cluster ions and all negative ions in WACCM.

(Jackman et al., 2011) reported the concentrations of OH, $HO_2$, $HNO_3$ and $O_3$ measured by the Microwave Limb Sounder (MLS) satellite instrument, and $NO_x$ data measured by the Advanced Composition Explorer (ACE). In the lower mesosphere (55 -70 km), $O_3$ decreased by 15-60% (with the largest change at ~12:00 UT on 17[th] January). Figure 3a shows that this behaviour is captured well by WACCM-rSIC, WACCM-D and WACCM-SIC, which all exhibit a decrease of 20-60% in only about 12 hours.

The polar NO, $NO_2$, OH, and $HO_2$ vertical profiles are shown in Figures 4-7. There is a marked increase in NO by a factor of 5 - 10 between 45 and 70 km, starting directly after the beginning of the SPE (02:10 UT on 16[th] January) and lasting for more than 15 days (Figure 4a). The NO then recovers back very slowly (~30-day timescale) to the original concentration.



The sudden increase of NO concentration after the SPE is picked up even by standard WACCM, indicating that this NO production is largely governed by the chemistry of the five major positive ions; however, standard WACCM predicts the NO enhancement down to 63 km, in contrast to the three SIC models where NO enhancement only occurs down to 70 km. This is most likely due to the significant role of negative ions in the 60-70 km region: the major negative ions, $O^-$, $CO_3^-$ and $ClO^-$,

convert NO to $NO_2$ or $NO_2^-$ (reactions NIR40, NIR57, NIR122 and NIR123 in the SM).

Figure 5a shows that the $NO_2$ density increases by a factor of ~3 after the beginning of the SPE. The effect is also seen in the standard WACCM run, but the effect is overestimated by a factor of 2.5 compared to WACCM-SIC. This is due to the lack of anion chemistry: $NO_2$ is converted to $NO_2^-$ via reaction with $O^-$, $O_2^-$ and $Cl^-$ (NIR4, NIR17, NIR113), or to $NO_3^-$ via reaction with $CO_3^-$ (NIR58 in the SM). Figure 5b shows that WACCM-D overestimates $NO_2$ on average by 40% compared to

WACCM-SIC and WACCM-rSIC. This appears to be due to the absence of $N_2$ clusters of proton hydrates and of the $O_2^+$ ion in WACCM-D. The relatively small difference between WACCM-rSIC and WACCM-SIC arises from the absence of the reactions of hydrated $O_2^-$ ions (NIR(28) and NIR(31) in the SM). In general, the pre-SPE NO and $NO_2$ concentrations from WACCM-D are more similar to WACCM than to WACCM-rSIC. Also, the post-SPE recovery of NO is more similar to the WACCM results, while the $NO_2$ recovery shows a mixed behaviour compared to WACCM-rSIC and WACCM: the increase

rate at the SPE is around half way compared to the WACCM-rSIC and WACCM results. These is explained by the different $NO_x$ chemistry in WACCM-D and WACCM-rSIC: as it can be seen in the SM, a the important recombination reactions of $NO^+$ clusters forming NO, namely RPE(8) and RPE(9) are missing in WACCM-rSIC. Although these do not include direct $NO_2$ formation, the increase in NO concentration will affect the $NO_2$ concentrations in both WACCM-D and WACCM-rSIC via the negative ion reactions of NIR(40), NIR(57).

Figure 6a shows that WACCM-SIC, -rSIC and -D all predict OH increases by a factor of ~5 between 53 and 70 km during the SPE, and a rapid recovery to quiet time levels after the SPE ends on January 23rd. In contrast, standard WACCM exhibits a modest enhancement of OH, and only above 63 km. Figure 7a shows $HO_2$ enhancements up to a factor of ~3 during the SPE, extending all the way down to 40 km - apart from standard WACCM. Again, the lack of anion chemistry seems to be responsible, since these $HO_x$ species are produced both by electron detachment (EDA(9), EDA(12) and EDA(13)) and anion-

molecule reactions (NIR(3), NIR(6), NIR(7), NIR(11), NIR(20), NIR(23), NIR(34), NIR(47), NIR(48), NIR(50), NIR(52), NIR(53), NIR(56), NIR(59), NIR(64), NIR(73), NIR(77) and NIR(80)). Figures 6b and 7b show that WACCM-rSIC produces generally better agreement with WACCM-SIC than WACCM-D. MLS observed a doubling of $HO_2$ between 70 and 80 km during the maximum of the SPE, and an even larger increase of up to 150% between 60 and 70 km (Jackman et al., 2011). In comparison, WACCM-SIC and -rSIC predict increases of 75% and 150% in these respective height ranges,

which are therefore in sensible agreement with the satellite observations. (Jackman et al., 2011) concluded that this large increase in $HO_2$ was mostly responsible for the catalytic destruction of $O_3$ in the mesosphere.

Figure 8a shows the $HNO_3$ density predicted by the four models. Standard WACCM is completely unable to reproduce the $HNO_3$ concentration enhancement predicted by WACCM-SIC, which underlines the importance of negative ion chemistry. The differences between WACCM-rSIC and WACCM-D are generally smaller than 50% and the effect of the SPE on $HNO_3$




is similar in both reduced models. Compared to WACCM-SIC, WACCM-rSIC overestimates $HNO_3$ by over a factor of 2 between 55 and 62 km during the SPE, while WACCM-D also overestimates $HNO_3$ but over a wide altitude range for a greater duration. The only $HNO_3$ consuming reactions that are eliminated in WACCM-rSIC are two fast $HNO_3$-cluster forming reactions (R1) [(NIR99)] and (R2) [(NIR104)]:

$NO_3^-(H_2O) + HNO_3 \rightarrow NO_3^-(HNO_3) + H_2O$ (R1)

$NO_3^-(HCl) + HNO_3 \rightarrow NO_3^-(HNO_3) + HCl$ (R2)

WACCM-SIC and WACCM-rSIC differ only in the ion chemistry. This indicates that the difference in the concentrations can arise only from this. Although (R1) and (R2) are present in WACCM-D, and therefore one would expect a smaller response of $HNO_3$ on the SPE, from the SM it is clear that there are a couple of indirect channels which increases the

concentration of $NO_3^-(H_2O)$ cluster (namely NIR(68) and NIR(91)); and $NO_3^-(H_2O)$ raises the $HNO_3$ concentration via NIR(97), which reaction also increases the $NO_3^-(HNO_3)$ concentration that raises the $HNO_3$ concentration via NIR(101). Therefore, the extra loss of $HNO_3$ via (R1) and (R2) are overcompensated by these $HNO_3$ increasing channels.

(Verronen et al., 2011) confirmed earlier work (Aikin, 1997; Verronen et al., 2008) which showed that the most important $HNO_3$-forming reactions in the mesosphere are the $H^+(H_2O)_n + NO_3^-(HNO_3)_m \rightarrow (m+1)HNO_3 + n\ H_2O$ ion-ion

recombination reactions. We found all these reactions to be necessary. However, the positive ion channels of direct $HNO_3$ formation are *all* redundant (i.e. (PIR97)-(PIR101) and (PIR102)-(PIR106) in the SM):

$H^+(H_2O)_n + N_2O_5 \rightarrow HNO_3 + H^+(H_2O)_{n-1}(HNO_3)$ (R3)

$H^+(H_2O)_n(HNO_3) + H_2O \rightarrow HNO_3 + H^+(H_2O)_{n+1}$ (R4)

The $NO_3^-(H_2O)_n + N_2O_5 \rightarrow HNO_3$ channel is also inefficient under all conditions. The $NO_3^-$ ion or its hydrates are

responsible for $HNO_3$ formation only via their reactions with proton hydrates, while $CO_3^-$, $O^-$, $O_2^-$ and $Cl^-$ are responsible for $HNO_3$ removal via reactions (R5)-(R8) [(NIR64), (NIR11), (NIR23), (NIR115)].

$CO_3^- + HNO_3 \rightarrow NO_3^- + OH + CO_2$ (R5)

$O^- + HNO_3 \rightarrow NO_3^- + OH$ (R6)

$O_2^- + HNO_3 \rightarrow NO_3^- + HO_2$ (R7)

$Cl^- + HNO_3 \rightarrow NO_3^- + HCl$ (R8)

The $NO_2^- \rightarrow NO_3^-$ conversion also removes $HNO_3$ by converting it into $HNO_2$ via (R9) [(NIR86)] below 80 km:

$NO_2^- + HNO_3 \rightarrow NO_3^- + HNO_2$ (R9)

Therefore, in WACCM-rSIC only the negative ion reactions affect the $HNO_3$ balance, as no positive ions contribute to its production or its loss. In contrast, the concentrations of $O_3$, $NO_x$ and $HO_x$ are influenced by both positive and negative ion

reactions which remain in the reduced model (see the SM for further details). It should be noted that the identification in the present study of unnecessary reactions of $N_2O_5$ allows a sub-mechanism to be extracted that still reproduces the mesospheric $O_3$, $HNO_3$, $HO_x$ and $NO_x$ profiles with reasonable accuracy.

A detailed analysis of the ionic reaction sequences and resulting changes in neutral species has been provided by (Verronen and Lehmann, 2013), who concluded that positive ion chemistry mainly leads to OH and NO formation, while negative ion



chemistry causes $HNO_3$ enhancement via $NO, NO_2, N_2O_5 \rightarrow HNO_3$ conversion. This is partly in accordance with our finding that the $HNO_3$ concentration is determined by negative ion reactions. However, we found that the OH concentration is also influenced by negative ion reactions of $O^-$, $OH^-$ and $CO_3^-$ (reactions PDE(4), NIR(3), NIR(7), NIR(11), NIR(47), NIR(48), NIR(52), NIR(53), NIR(59), NIR(64) and NIR(73) in the SM). According to Verronen and Lehmann, mesospheric NO

production is dominated by positive ion reactions; here we conclude that NO is consumed mostly by negative ion reactions.

## 4 Conclusions

In this paper we have described the development of a whole *D* region chemistry based on the Sodankylä Ion and Neutral Chemistry (SIC) coupled into WACCM. Systematic mechanism reduction using the Simulation Error Minimization Connectivity method produces a *D* region ion chemistry described by 181 ion-molecule reactions, a 41% reduction from the

full SIC chemistry. Our earlier WACCM-D version has 196 reactions; the majority of the reactions in WACCM-rSIC and WACCM-D are the same. However, WACCM-D does not include reactions of the larger proton hydrates and their clusters, as well as the reactions of $O_2^+$ clusters, and inclusion of these reactions gives better agreement with the $O_3$ predicted by the full WACCM-SIC chemistry. WACCM-rSIC runs 25% faster than WACCM-SIC, which is a reasonable improvement. Moreover, the simulation time with WACCM-rSIC is only 90% more than with the standard version of WACCM, which

contains no negative ions or positive cluster ions and is clearly inadequate for simulating the impacts of energetic particle precipitation on the chemistry of the middle atmosphere.

## Acknowledgements

The authors thank the UK Natural Environmental Research Council (NERC) for funding (standard grants NE/J02077X/1 and NE/J022187/1). Dr Kevin Hughes (University of Sheffield) and Prof Tamás Turányi (Eötvös Loránd University, Budapest)

are thanked for helpful discussions. The work of PTV and MEA was supported by the Academy of Finland through the project #276926 (SECTIC: Sun-Earth Connection Through Ion Chemistry). NCAR is funded by the National Science Foundation and DRM is supported in part by NASA Living With a Star Grant NNX14AH54G.

## Code and data availability

WACCM (CESM) 1.1.1 version is available from  https://svn-ccsm-release.cgd.ucar.edu/model_versions/cesm1_1_1

The combined WACCM-SIC and the reduced version, WACCM-rSIC as well as all model simulation results are available upon request to J.M.C.P while the WACCM-D and the SIC model are available  upon request to P.T.V.



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





**Table 1**. Species of the rSIC model. The set of important species for which the concentrations are required to be reproduced within 5% RMS is shown in bold type; the rest are necessary species.

| Neutrals | **$HNO_3$, $O_3$, $H_2O_2$, NO, $NO_2$, $HO_2$, OH, $N_2O_5$** <br> $CH_2O$, $CH_3$, $CH_4$, Cl, HCl, N, ClO, $N(^2D)$, $HNO_2$, $N_2$, CO, $CO_2$, $N_2O$, $CO_3$, O, H, $O(^1D)$, $H_2$, $NO_3$, $O_2$, $H_2O$, $O_2(^1\Delta_g)$ |
|---|---|
| Cations | **$O_2^+$, $O_4^+$, $NO^+$, $NO^+(H_2O)$, $O_2^+(H_2O)$, $H^+(H_2O)$, $H^+(H_2O)_2$, $H^+(H_2O)_3$, $H^+(H_2O)_4$** <br> $H^+(H_2O)_5$, $H^+(H_2O)_6$, $H_3O^+(H_2O)_2(CO_2)$, $H_3O^+(OH)$, $O_2^+(CO_2)$, $H_3O^+(OH)(CO_2)$, $H_3O^+(OH)(H_2O)$, $O_2^+(H_2O)(CO_2)$, $O_2^+(H_2O)_2$, $O_2^+(N_2)$, $NO^+(H_2O)_2$, $H^+(H_2O)(CO_2)$, $O^+$, $N^+$, $N_2^+$, $NO^+(H_2O)_3$, $O_4^+$, $H^+(H_2O)_2(CO_2)$, $H^+(H_2O)_2(N_2)$ |
| Anions | **$O_3^-$, $O^-$, $O_2^-$, $OH^-$, $O_2^-(H_2O)$, $O_2^-(H_2O)_2$, $O_4^-$, $CO_3^-$, $CO_3^-(H_2O)$, $CO_4^-$, $HCO_3^-$, $NO_2^-$, $NO_3^-$, $NO_3^-(H_2O)$, $NO_3^-(H_2O)_2$, $NO_3^-(HNO_3)$, $NO_3^-(HNO_3)_2$, $Cl^-$, $ClO^-$** <br> $NO_2^-(H_2O)$, $Cl^-(H_2O)$, $Cl^-(CO_2)$, $Cl^-(HCl)$ |




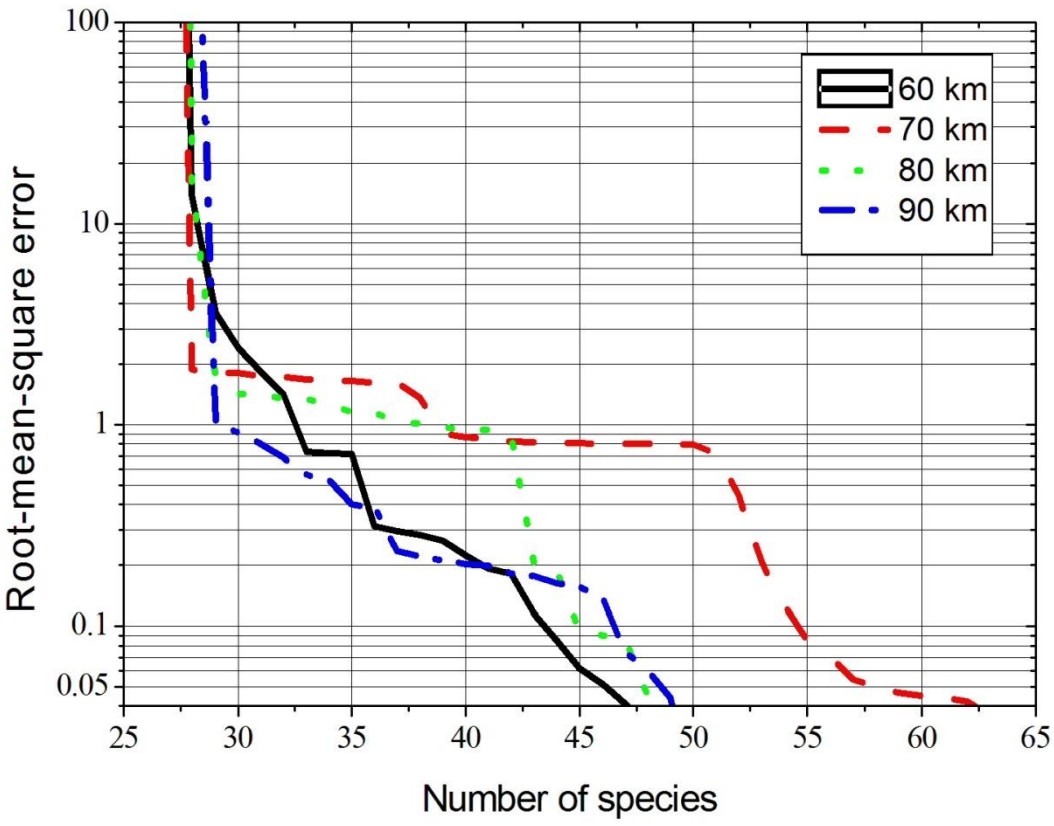

**Figure 1.** Root-mean-square (rms) error as a function of number of species in the reduced mechanism for the four selected altitudes (60, 70, 80 and 90 km).









**Figure 2. Atmospheric concentration profiles and relative differences calculated using the full SIC model and reduced (rSIC) model for 17th January 2005 at 17:50 UT. In the left-hand panels the solid lines show the concentrations calculated by the full SIC model, while the symbols refer to the reduced rSIC model. The right-hand panels show the percentage difference between the rSIC and SIC models: (a) concentrations and (b) percentage differences of $HNO_3$, $NO$, $NO_2$, $O_3$ and $H_2O_2$; (c) concentrations and (d) percentage differences of $OH$ and $HO_2$; (e) concentrations and (f) percentage differences of $NO^+$ and $O_2^+$; (g) concentrations and (h) percentage differences of $CO_3^-$, $NO_3^-$ and $O_2^-$.**



**(a)**

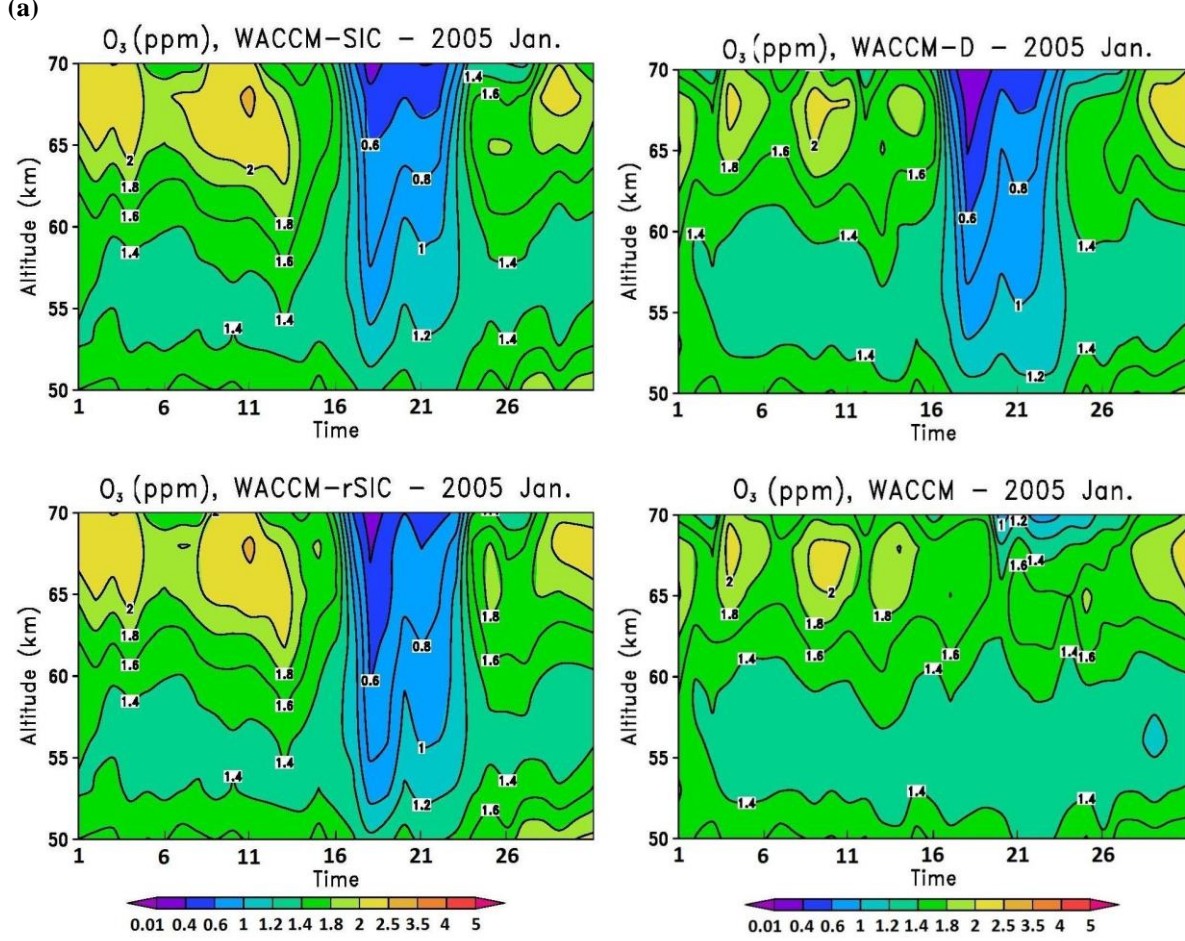





**(b)**

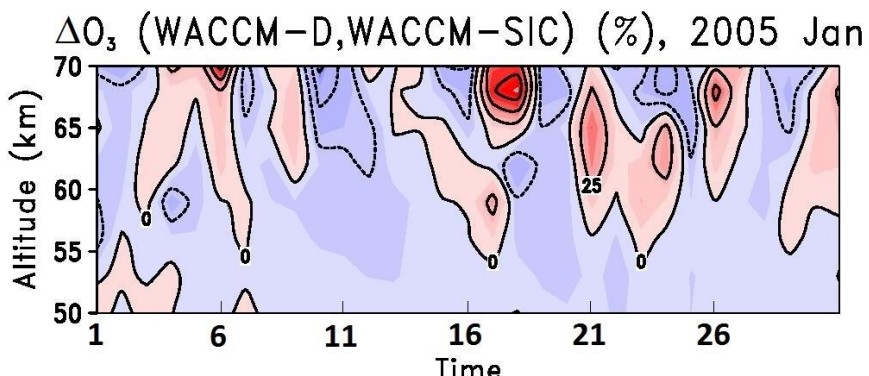

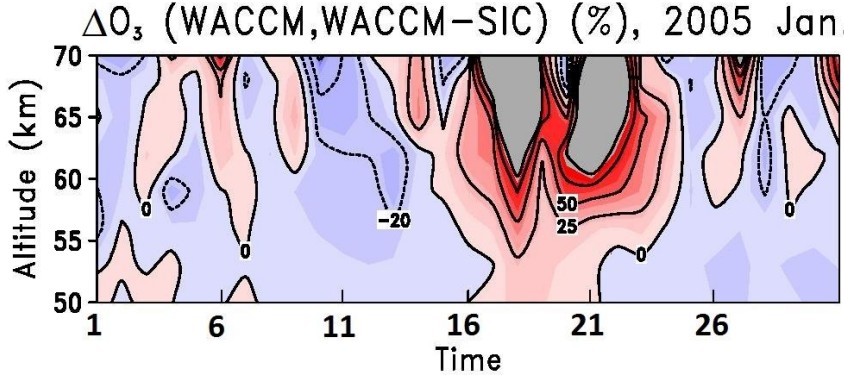

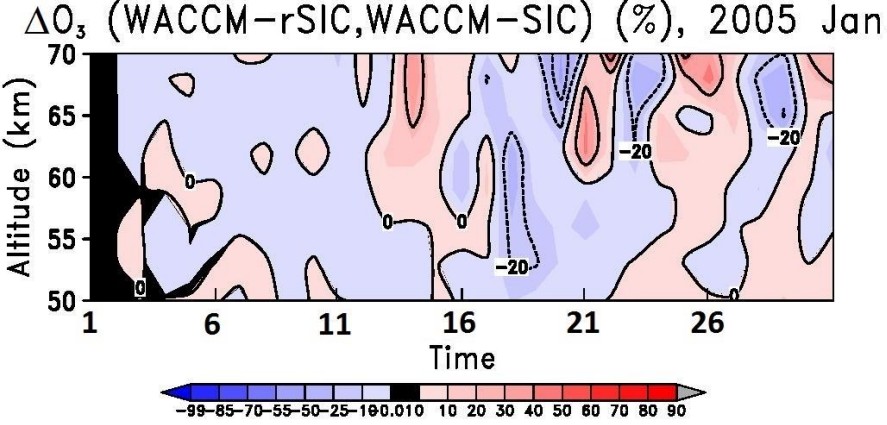





**Figure 3. (a) Zonal mean vertical concentration profiles of mesospheric O$_3$ (in ppm) for northern hemisphere polar latitudes (60°-90°N) during January 2005; (b) percentage difference between three models (WACCM-D, WACCM and WACCM-rSIC) and the full WACCM-SIC.**



**(a)**

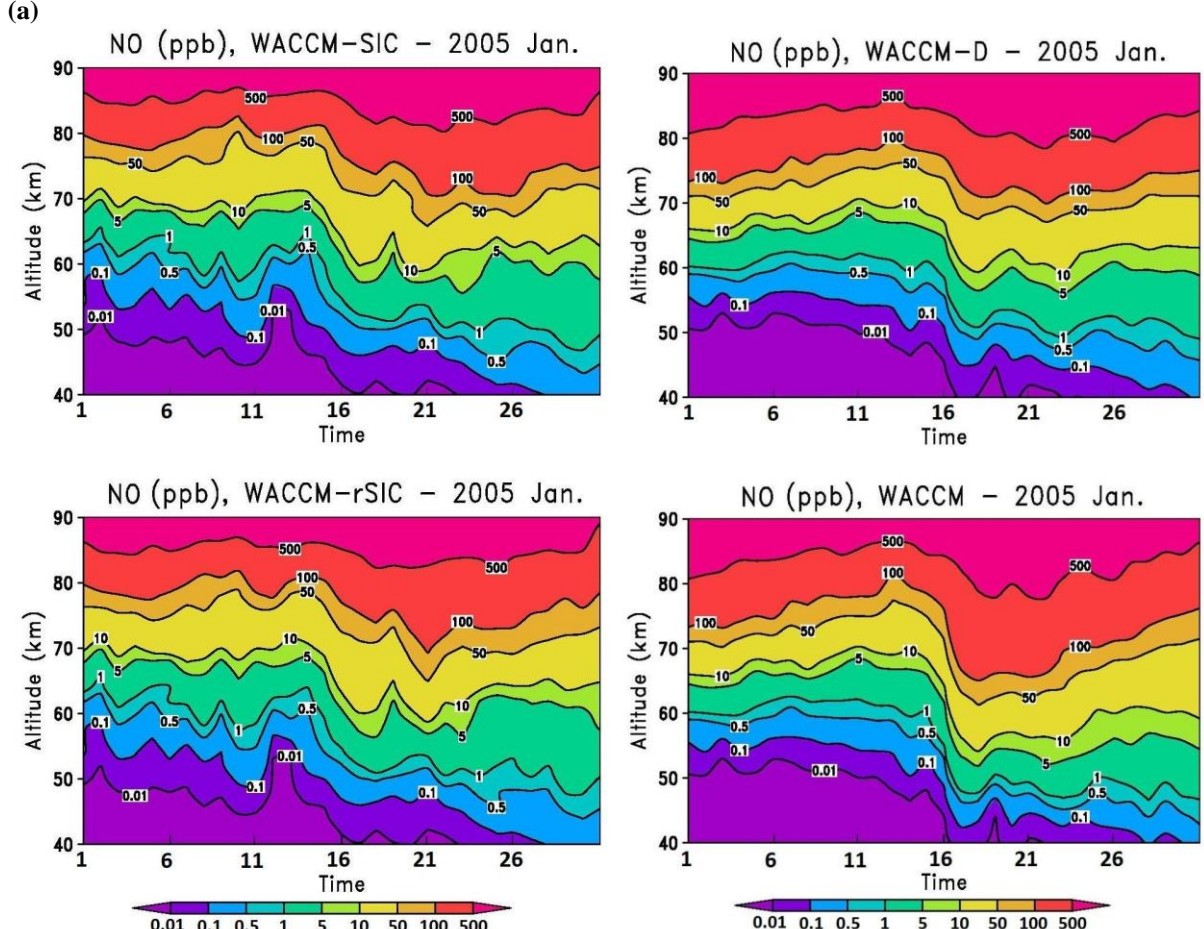

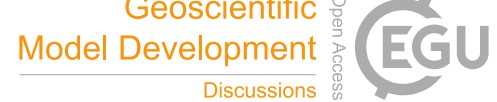



**(b)**

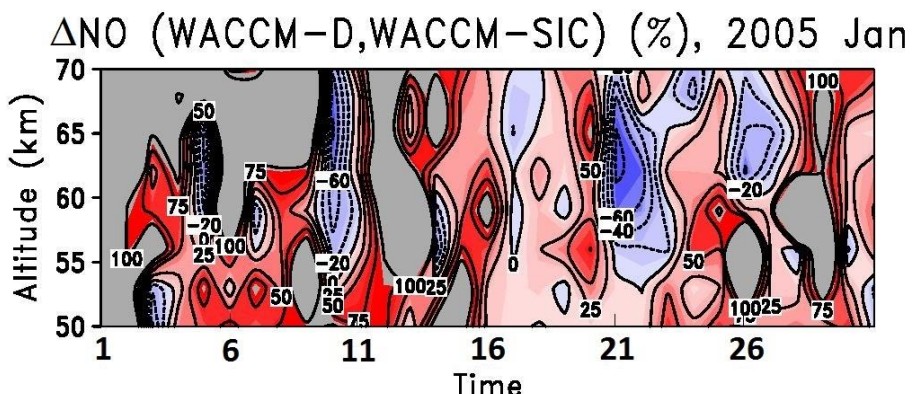

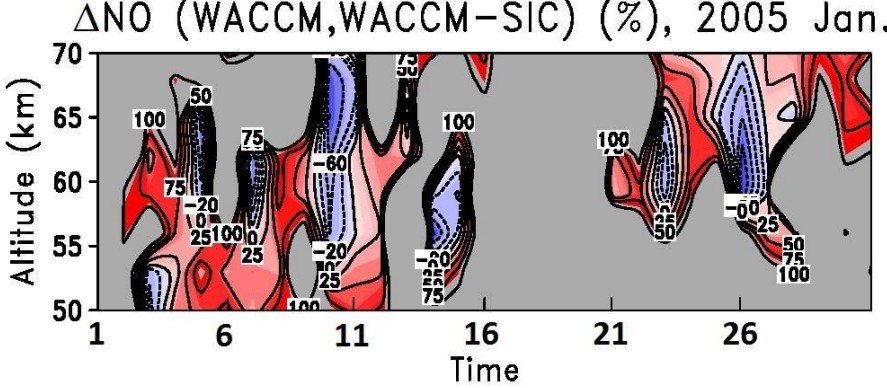

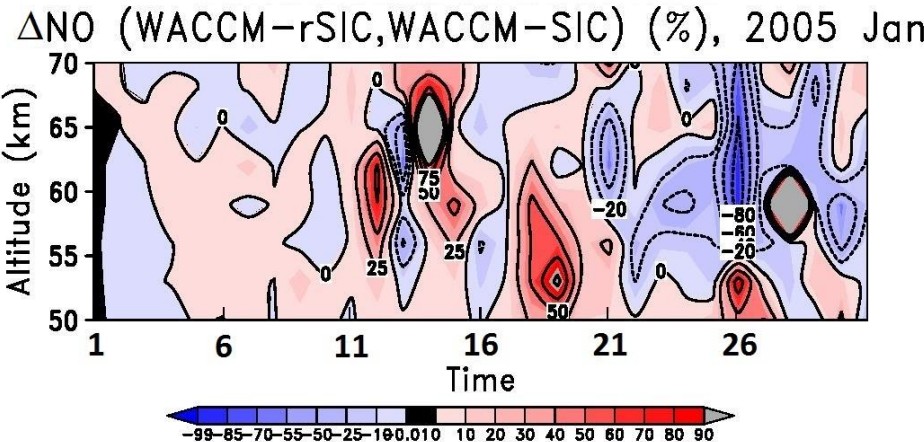



**Figure 4. (a) Zonal mean vertical concentration profiles of mesospheric/upper stratospheric NO (in ppb) for northern hemisphere polar latitudes (60°-90°N) during January 2005; (b) percentage difference between three models (WACCM-D, WACCM and WACCM-rSIC) and the full WACCM-SIC.**



**(a)**

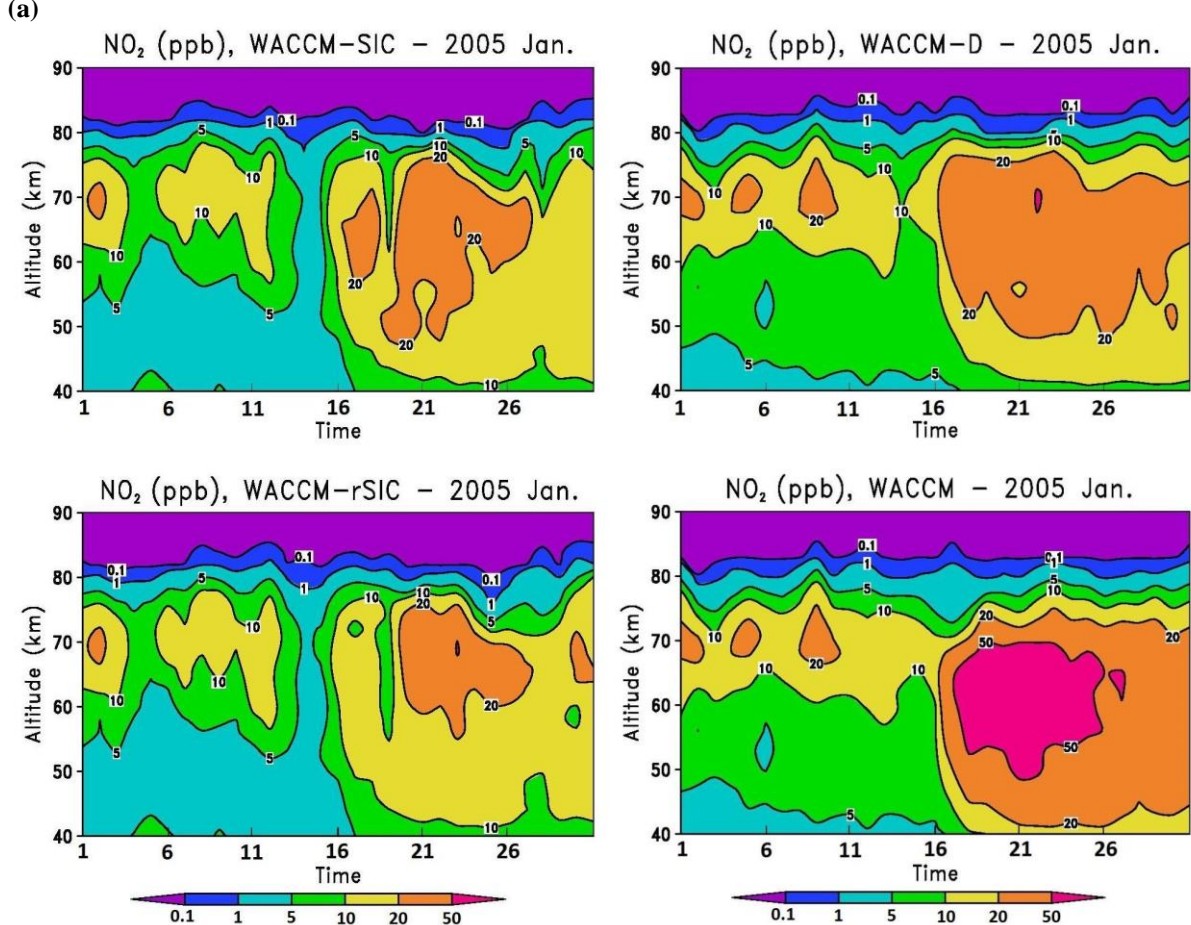



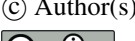

**(b)**

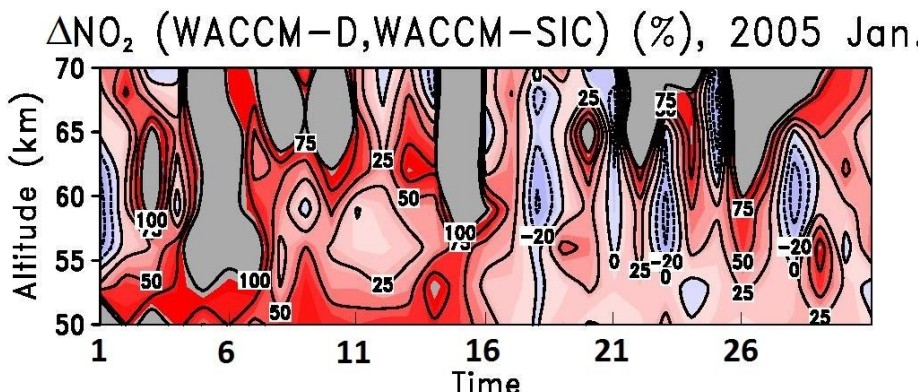

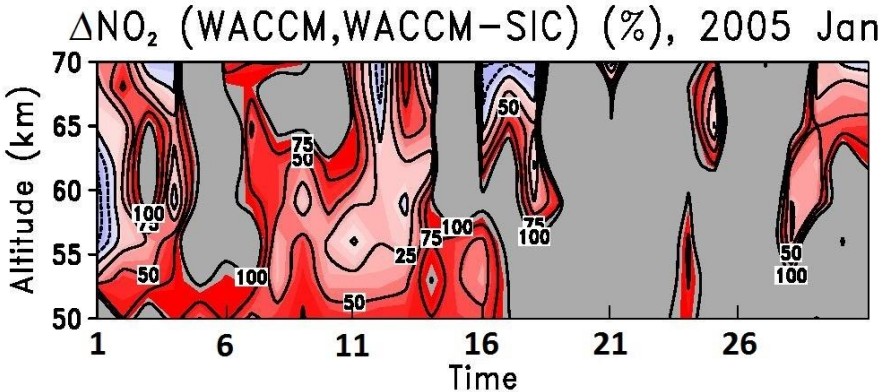

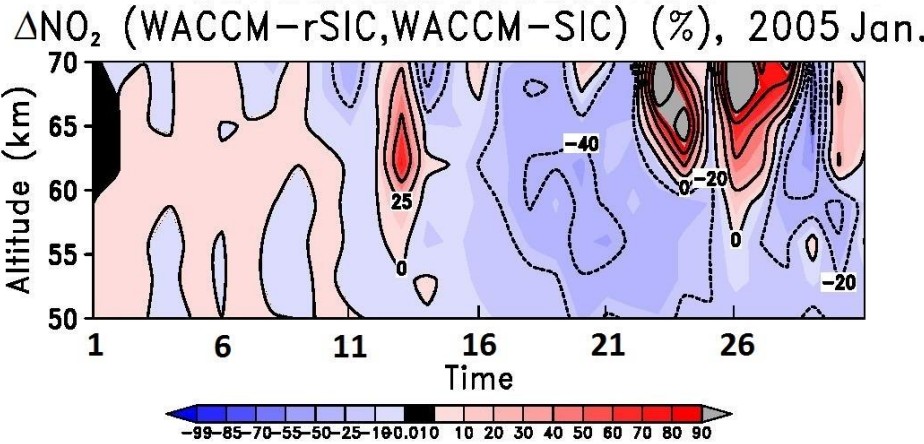



**Figure 5. (a) Zonal mean vertical concentration profiles of mesospheric/upper stratospheric NO (in pp) for northern hemisphere polar latitudes (60°-90°N) during January 2005; (b) percentage difference between three models (WACCM-D, WACCM and WACCM-rSIC) and the full WACCM-SIC.**



**(a)**

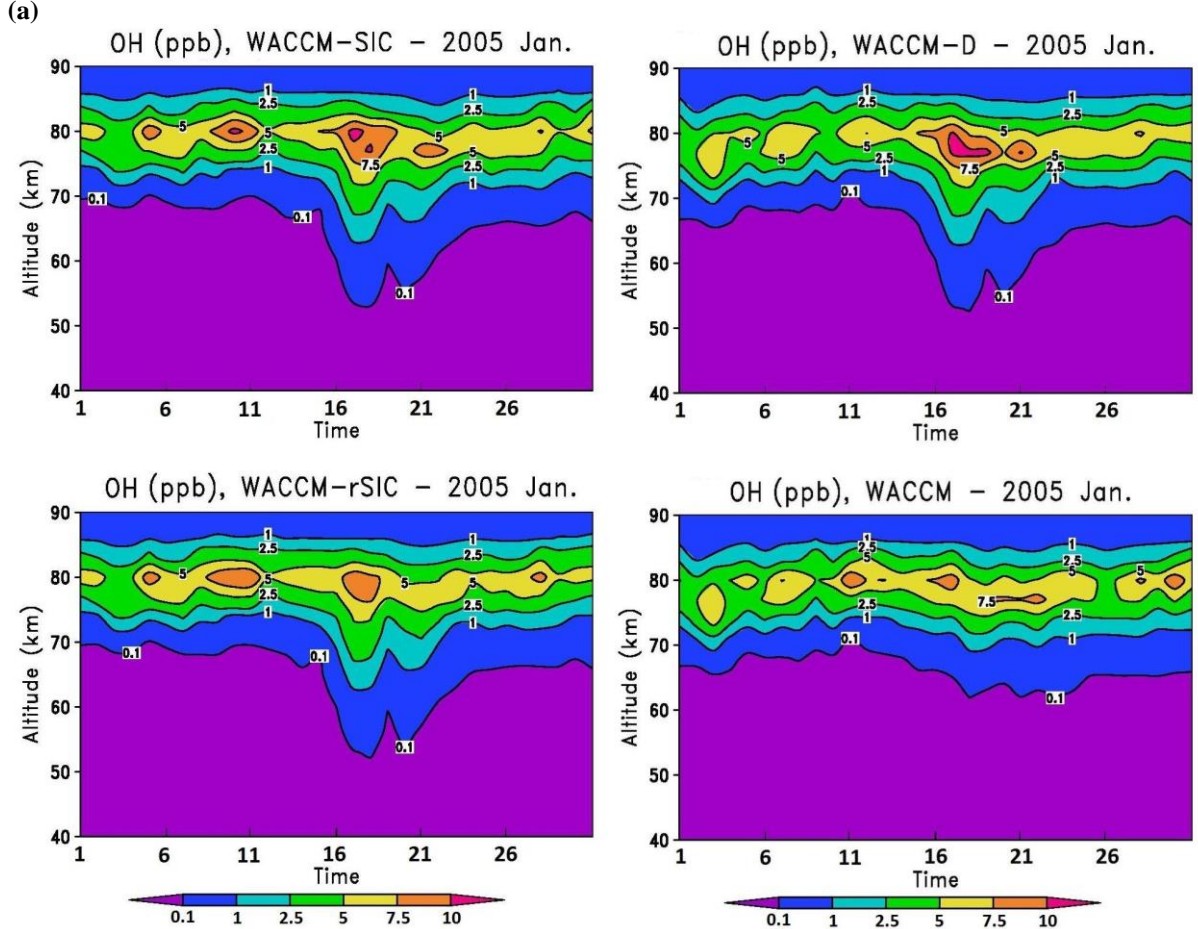





**(b)**

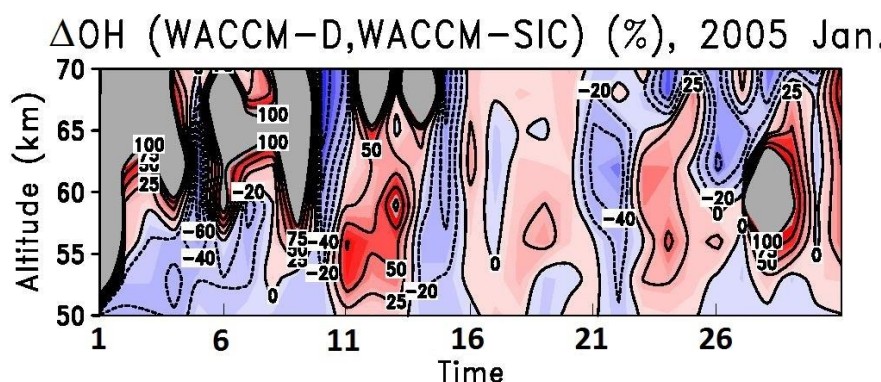

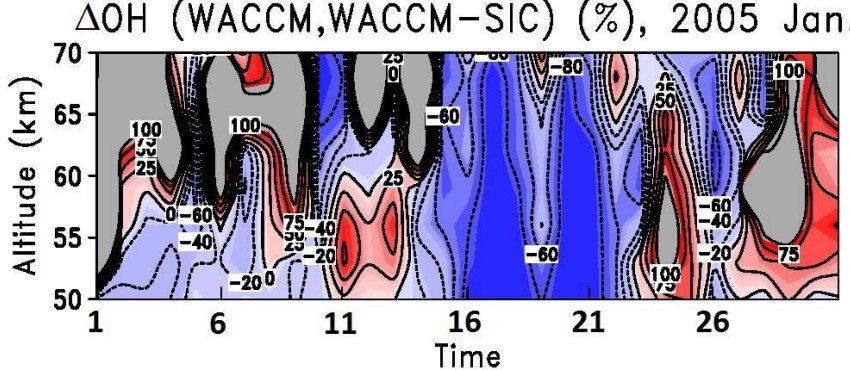

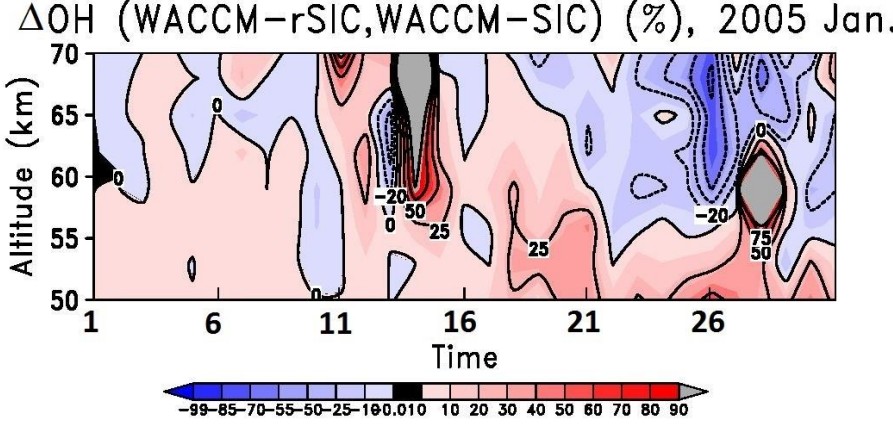





**Figure 6. (a) Zonal mean vertical concentration profiles of mesospheric/upper stratospheric OH (in ppb) for northern hemisphere polar latitudes (60°-90°N) during January 2005; (b) percentage difference between three models (WACCM-D, WACCM and WACCM-rSIC) and the full WACCM-SIC.**





**(a)**

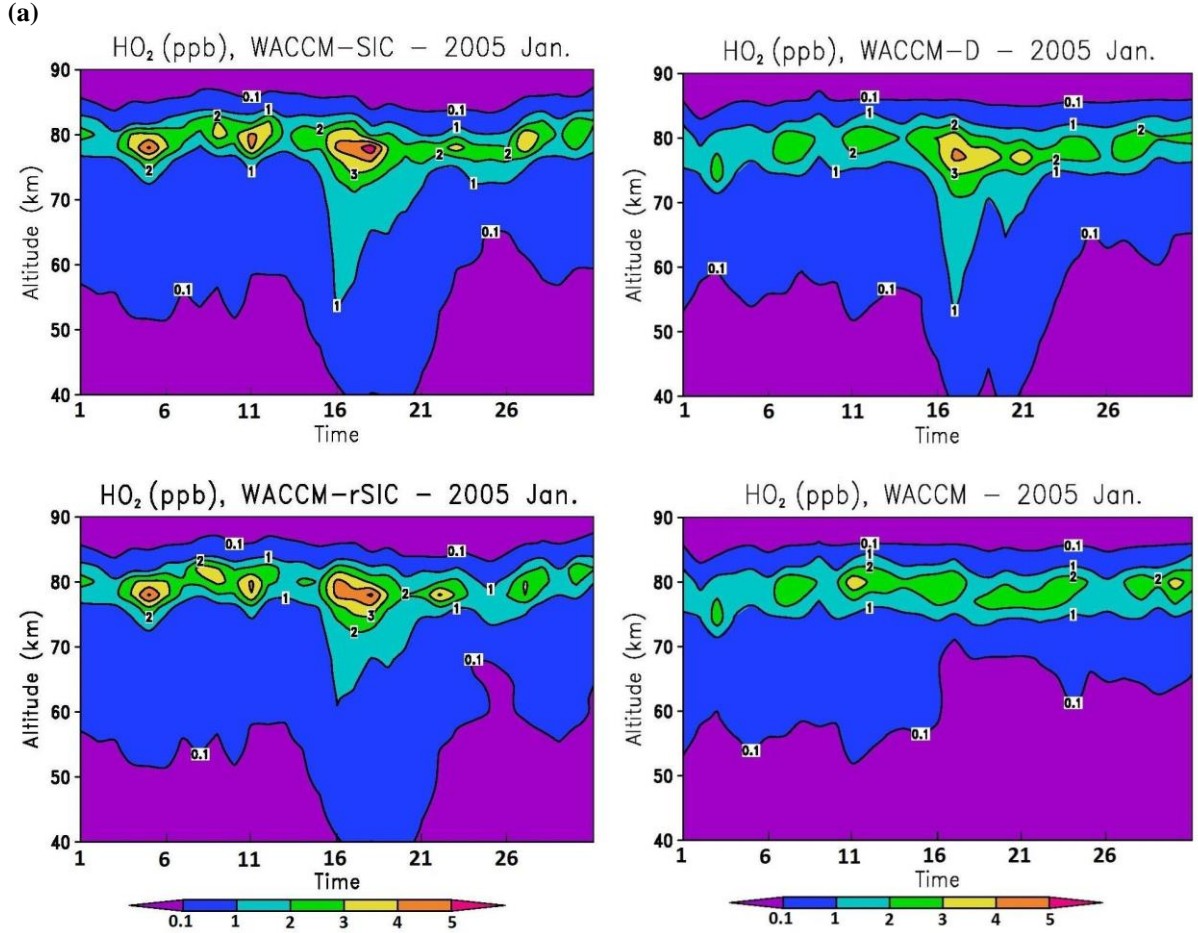





**(b)**

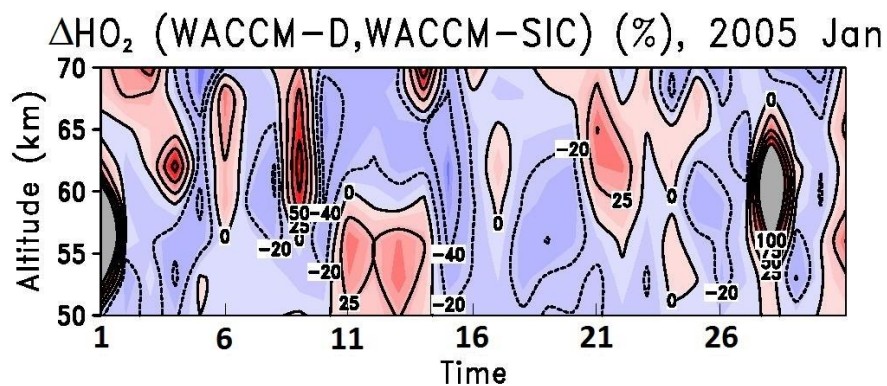

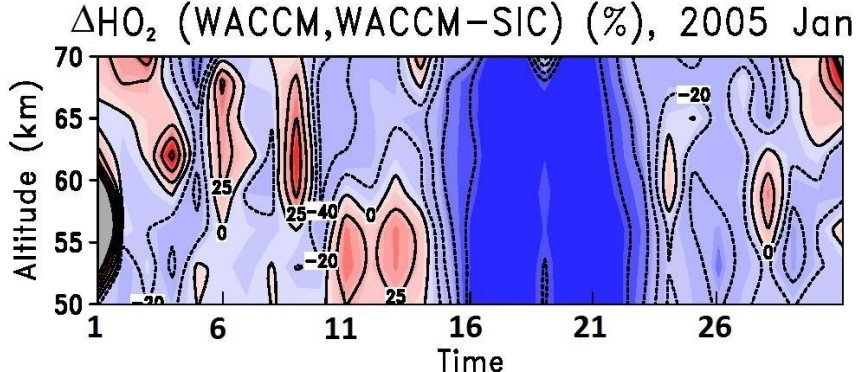

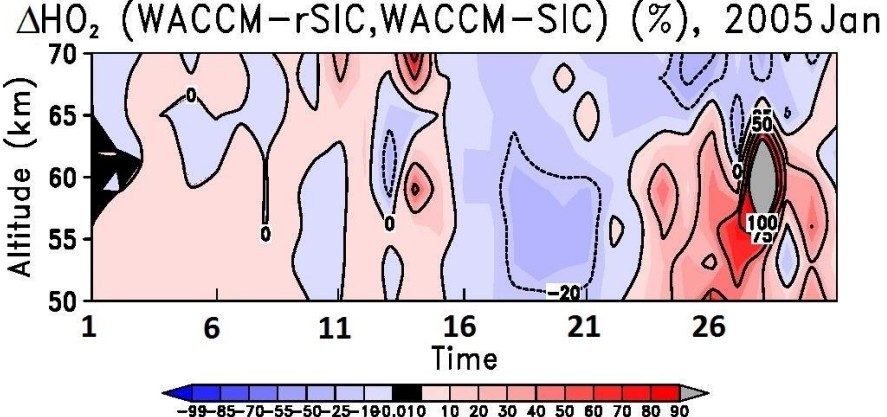





**Figure 7. (a) Zonal mean vertical concentration profiles of mesospheric/upper stratospheric HO$_2$ (in ppb) for northern hemisphere polar latitudes (60°-90°N) during January 2005; (b) percentage difference between three models (WACCM-D, WACCM and WACCM-rSIC) and the full WACCM-SIC.**





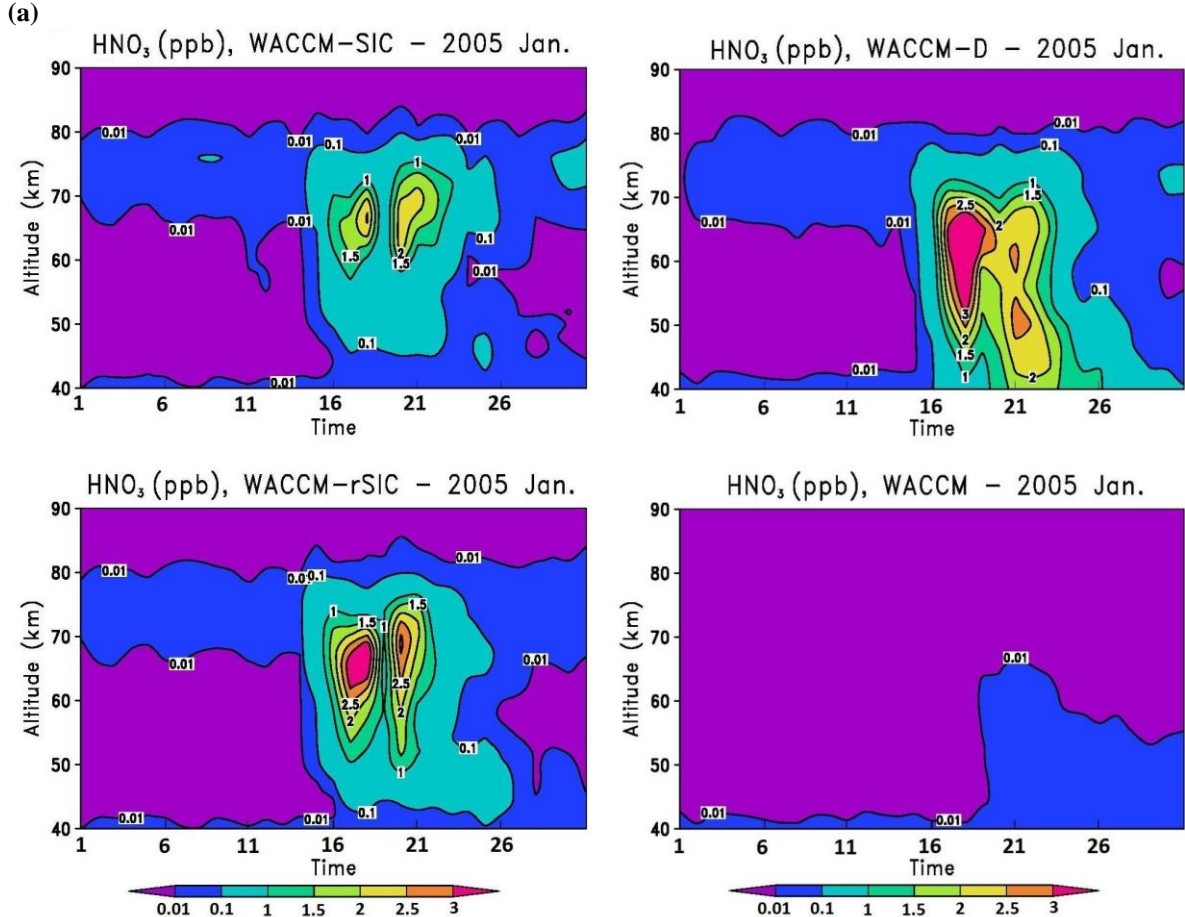

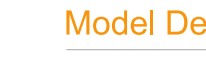
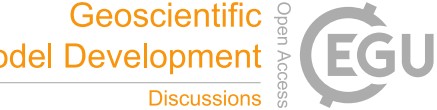

**(b)**

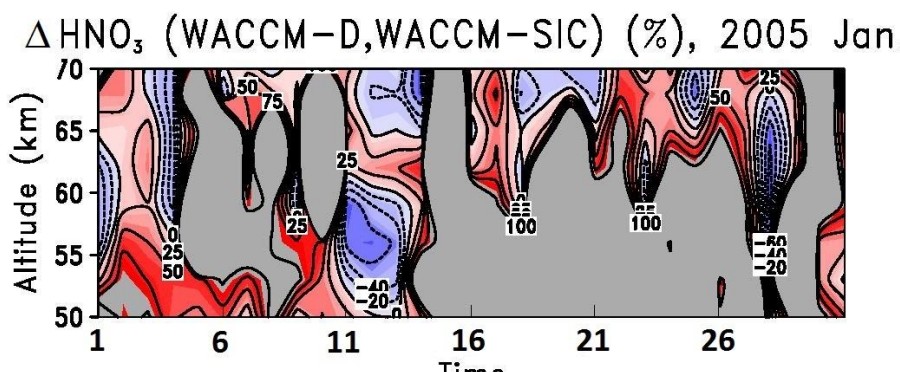

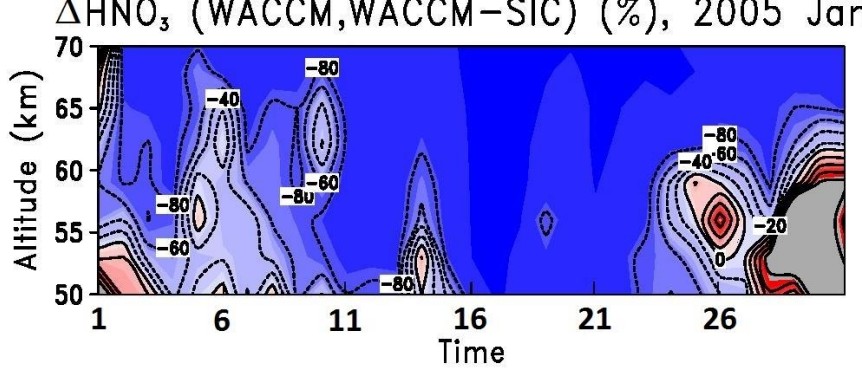

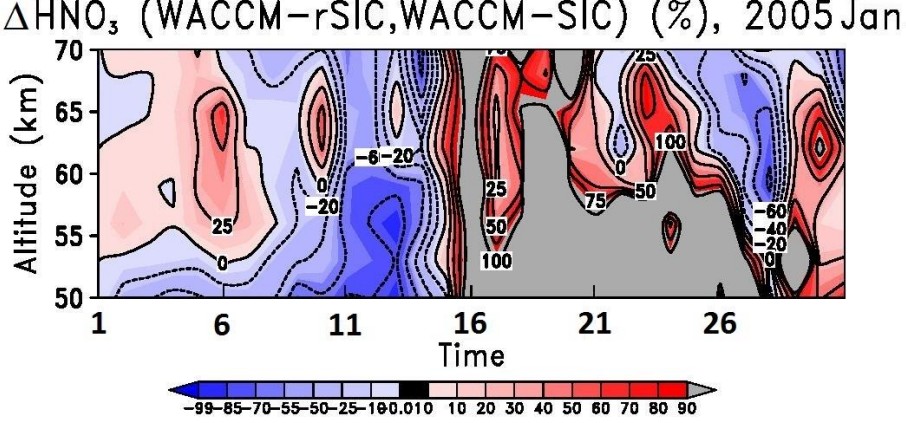



**Figure 8. (a) Zonal mean vertical concentration profiles of mesospheric/upper stratospheric HNO₃ (in ppb) for northern hemisphere polar latitudes (60°-90°N) during January 2005; (b) percentage difference between three models (WACCM-D, WACCM and WACCM-rSIC) and the full WACCM-SIC.**