# Peer review of "D* region ion-neutral coupled chemistry (Sodankylä Ion Chemistry, SIC) within a Whole Atmosphere Chemistry-Climate Model (WACCM 4) – WACCM-SIC and WACCM-rSIC"

_Geoscientific Model Development, 2016_

## Short Comment (SC1) · 20 Apr 2016

Dear authors,

In my role as Executive editor of GMD, I would like to bring to your attention our Editorial version 1.1:

http://www.geosci-model-dev.net/8/3487/2015/gmd-8-3487-2015.html

This highlights some requirements of papers published in GMD, which is also available on the GMD website in the 'Manuscript Types' section:

http://www.geoscientific-model-development.net/submission/manuscript_types.html

In particular, please note that for your paper, the following requirement has not been met in the Discussions paper:

[Figure]

- "The main paper must give the model name and version number (or other unique identifier) in the title."

Please add a version number or other unique identifier for your model in the title upon your revised submission to GMD. Best of all you should provide version numbers for the WACCM model as well as for your own developments which are, if I understand correctly, not part of the official model version.

Additionally, I ask you to move the Code availability section in front of the Acknowledgements.

Yours,

Astrid Kerkweg

―――――――――――――――――――――

---

## Short Comment (SC2) · 20 Apr 2016

Dear Dr Kerkweg,

Thank you for your comments.

The title will be changed to: "D region ion-neutral coupled chemistry (Sodankylä Ion Chemistry, SIC) within a Whole Atmosphere Chemistry-Climate Model (WACCM 4) – WACCM-SIC and WACCM-rSIC"

The Code availability section will be moved in front of the Acknowledgements.

Kind regards, Tamas Kovacs
* * *

---

## Referee Comment (RC1) · Anonymous Referee #1 · 26 May 2016

General Comments

This is a generally clear, well written paper. The work is of excellent scientific quality, is reproducible and is of high scientific significance:

Specific Comments

p7, line 5 - a 5 year spin up period seems very long - similar dynamics-only experiments would likely require a considerably shorter spin up. Is the longer spin up chosen because of the very long lifetime of some of the key chemical reactions? This should be explained in the text - others may want to extend these experiments to other time periods and other events and will want to know (for reasons of computational efficiency) whether they need such a long spin up each time.

p12 Conclusions - given that the rSIC approach was trained using one period of data, and tested using just one period the authors should comment more on the wider applicability of the approach

Technical Corrections

p2, line 22 - this states the SIC model has 306 ion-molecule reactions but 307 are listed in the SM

p2, line 29 - WACCM-D, WACCM-SIC, etc are defined in the abstract but I think they should be defined again in the main text.

p4, line 12 - From the SM it is not clear which reactions are the 7 photoionization and 16 photodissociation reactions mentioned in the text (though some will be the ones marked PDE*) - this should be clarified.

p8, line 2, and Figure 2 - the legend suggests the profiles from two WACCM experiments are plotted on the left hand panels, but the results appear identical and do not appear to correspond with the relative differences plotted on the right hand panels. Either alter the left hand panels to show the differences better or (if the log scaling and rage of values makes this impossible) indicate in the text that the differences can only be clearly seen n the right hand panels to clarify and to avoid the reader looking for a needle in a haystack on the left.

---

## Referee Comment (RC2) · Anonymous Referee #2 · 6 Jul 2016

**1 General comments**

This paper is the second dealing with including a simplified D-region ion chemistry scheme in the WACCM model using a subset of SIC ion chemistry reactions. Compared to the development of WACCM-D discussed in the previous paper (Verronen et al., JGR, 2016), in this paper a more systematic approach is used in the reduction of the ion-chemistry reactions. WACCM-D already showed a significant improvement in some neutral mesospheric species compared to model versions without, or with parametrized, ion chemistry. This paper clearly demonstrates that using a systematic approach of reaction reduction yields a better reproduction of the key neutral species (NOx, HOx, ozone, HNO3) during large ionization events in the mesosphere than WACCM-D, while at the same time reducing the number of ion reactions implemented even more. Both the description of the reduction process, as well as the list of full and reduced ion chemistry in the supplementary material, are of great interest to other modelers achieving similar. The paper is generally clearly and well written.

**2 Specific comments**

I was a bit surprised that for the comparison with standard WACCM, you include a parametrization for NOx production, but not for the production of HOx (page 6, lines 21-23). A parametrization for HOx production due to ionization based on positive ion chemistry has been developed already by Solomon et al. (1981). This parametrization is widely used in models to study the impact of solar proton events and middle-energy electron events on the stratosphere and mesosphere (e.g. Jackman et al. 2000, 2001, 2005a, b; Rohen et al. 2005; Funke et al., 2011). It has also been included in WACCM to study solar proton events (e.g., Jackman et al., 2011). Why not use this version here as standard WACCM? The Solomon parametrization is based on HOx production by positive protonized water cluster ions, an idea already developed in the 1970th and 1980th (Swider and Keneshea, 1973; Swider et al., 1978; Crutzen and Solomon, 1980; Solomon et al., 1981). It has been shown in a number of studies since then that this parametrization generally leads to a reasonable representation of HOx production and ozone loss (see, e.g., Jackman et al. 2001; Funke et al. 2011). That a model not using a parametrization of the HOx production by protonized water cluster ions underestimates HOx production and subsequent mesospheric ozone loss during large solar proton events has been shown in a number of publications since the 1970th (see also page 9, line 23-24). However, I would assume that the reduced ion chemistry provides a better representation of the full ion chemistry than the Solomon parametrization which simplifies, e.g., the vertical range of HOx production which depends on the availability of water vapor.
Page 7, line 13 ff, discussion of Figure 1: is this the change of rms error of one species, or of all species considered? If the first, which? What are the units on the y-axis,

Page 10-19: I found the discussion in this paragraph confusing. In particular you mention several times that "important reactions" are missing in the reduced SIC scheme as the reason for a disagreement between WACCM-rSIC and WACCM-D which apparently includes these "important" reactions. However, at the same time, results from the reduced scheme agree better with results using the full ion chemistry scheme than WACCM-D results (e.g., NO and NO2 before and after the solar proton event: WACCrSIC really is in much better agreement with WACCM-SIC than WACCM-D there). This suggests that the reactions included in WACCM-D but not in WACCM-rSIC are actually not important, and should not be called such without further justification (e.g., reactions of hydrated O2- ions in line 11/12, and NO+ clusters in line 17).

**3 Technical comments**

Abstract, page 1, line 17: could you include the number of ion species in the reduced SIC scheme as well?

Introduction, page 2, line 4, 12, and 17: the reference is Sinnhuber et al., 2012, or Sinnhuber, Nieder and Wieters, 2012.

Introduction, page 2, line 20: SIC is certainly "one of the" leading kinetic models of D region chemistry, though not "the" leading kinetic model of D region chemistry.

Methodology, page 3, line 6, caption of table in supplementary material: do you mean the "red" bold type? Please also make a note in the caption of the table.

Methodology, page 3, line 10-11: Primary protons as well as secondary electrons will also collide with O2, leading to the same reactions as with N2, namely ionization, dissociation, and excitation (Porter et al., 1976). Also, ionization of O should play a
role in the upper mesosphere and lower thermosphere, where O becomes one of the most abundant species. Is this not included in SIC? Please clarify.

Methodology, pare 4, line 1: does the reduced mechanism contain all reactions of the important + necessary species, or a subset? E.g., is there an attempt to reduce/limit the number of reactions of the species included?

Page 8, line 7,  $\dots$  neutral profiles are reproduced within a factor of 2  $\dots$  in Figure 2, all neutral profiles are within the 5

Page 9, line 23/24: that models not including the effect of positive water cluster ions underestimate the HOx production and ozone loss during solar proton events has been discussed since the 1970th, see specific comment above, and (Jackman et al., GRL, 2001) for one example.

Page 10, line 1-2: Earlier (line 6, line 21-22) you mention that the production of NOx due to ionization in the standard WACCM is parametrized to 1.25 NOx per ion pair; here you suggest that the NOx production is due to the five-ion chemistry scheme implemented in standard WACCM (for the lower thermosphere). Please clarify which is correct.

Conclusions, page 12, line 13: You could include an additional statement like: Before and after the solar proton event, NO and NO2 from WACCM-rSIC agree much better with results from WACCM-SIC than the results of WACCM-D, because ....

---

## Author Response (AR1)

Answers to Reviewers' Comments on Kovacs *et al.*,
*D* region ion-neutral coupled chemistry within a whole atmosphere chemistry-climate model

**Reviewer 1**

Comment 1: p7, line 5 - a 5 year spin up period seems very long - similar dynamics-only experiments would likely require a considerably shorter spin up. Is the longer spin up chosen because of the very long lifetime of some of the key chemical reactions? This should be explained in the text

**Answer 1: Explanation is given in the text. "The long spin up period was used in order to definitely allow enough time to reach steady-state and also to count for long lived species such as $N_2O$, $CH_4$." (page 7, line 18-19)**

Comment 2: p12 Conclusions - given that the rSIC approach was trained using one period of data, and tested using just one period the authors should comment more on the wider applicability of the approach.

**Answer 2: Comment is given in the text. "Note that the rSIC model contains temperature (and pressure) dependent rate coefficients (where available), therefore the chemical model can be used over a wide range of conditions." (page 12, lines 30-31)**

Comment 3: p2, line 22 - this states the SIC model has 306 ion-molecule reactions but 307 are listed in the SM

**Answer 3: Corrected in the text; 306 changed to 307. (page 1, line 16; page 2, line 22; page 3, line 5; page 4, line 15; page 8, line 8)**

Comment 4: p2, line 29 - WACCM-D, WACCM-SIC, etc are defined in the abstract but I think they should be defined again in the main text.

**Answer 4: Corrections made in the text. (page 2, line 29)**

Comment 5: p4, line 12 - From the SM it is not clear which reactions are the 7 photoionization and 16 photodissociation reactions mentioned in the text (though some will be the ones marked PDE*) - this should be clarified.

**Answer 5: Photoionization and photodissociation reactions are not listed in the SM as no fixed rate parameters are defined for them (they depend on the solar activity). They are listed separately in Table 2. This is made clear in the text (page 4, lines 15-17).**

Comment 6: p8, line 2, and Figure 2 - the legend suggests the profiles from two WACCM experiments are plotted on the left hand panels, but the results appear identical and do not appear to correspond with the relative differences plotted on the right hand panels. Either alter the left hand panels to show the differences better or (if the log scaling and rage of values makes this impossible) indicate in the text that the differences can only be clearly seen n the right hand panels to clarify and to avoid the reader looking for a needle in a haystack on the left.

**Answer 6: Corrections and clarification made in the text. It is not possible to show the differences better on the left panels. "Note that the differences can only be seen in the right hand panels of Figure 2." (page 8, lines 17-18)**

Comment 1: A parametrization for HOx production due to ionization based on positive ion chemistry has been developed already by Solomon et al. (1981)… Why not use this version here as standard WACCM?

**Answer 1: This is corrected and clarified in the text: "The $HO_x$ species are produced via a complicated ion chemistry scheme that is based on (Solomon *et al.*, 1981)" (page 6, lines 30-32)**

Comment 2: discussion of Figure 1: is this the change of rms error of one species, or of all species considered? If the first, which? What are the units on the y-axis?

**Answer 2: Figure legend is corrected and clarified. RMS is for all species considered and they are dimensionless. (Figure 1 legend, page 20, line 4)**

Comment 3: Results from the reduced scheme agree better with results using the full ion chemistry scheme than WACCM-D results… This suggests that the reactions included in WACCM-D but not in WACCM-rSIC are actually not important, and should not be called such without further justification.

**Answer 3: The authors agree with this comment and the "important" term has been deleted from the sentence "as it can be seen in the SM, the  recombination reactions of $NO^+$ clusters forming NO". (page 10, lines 26-27)**

Comment 4: Abstract, page 1, line 17: could you include the number of ion species in the reduced SIC scheme as well?

**Answer 4: It is clarified now. "…a reaction scheme of 181 ion-molecule reactions of 27 positive and 18 negative ions" (page 1, lines 18-19)**

Comment 5: Introduction, page 2, line 4, 12, and 17: the reference is Sinnhuber et al., 2012, or Sinnhuber, Nieder and Wieters, 2012.

**Answer 5: Reference is corrected in the text. (page 2, lines 5, 12, 17)**

Comment 6: Introduction, page 2, line 20: SIC is certainly "one of the" leading kinetic models of D region chemistry, though not "the" leading kinetic model of D region chemistry.

**Answer 6: Corrected in the text. (page 2, line 20)**

Comment 7: Methodology, page 3, line 6, caption of table in supplementary material: do you mean the "red" bold type? Please also make a note in the caption of the table.

**Answer 7: Yes, it is the "red bold type". Corrected in the text. (page 3, line 7)**

Comment 8: Methodology, page 3, line 10-11: Primary protons as well as secondary electrons will also collide with O2, leading to the same reactions as with N2, namely ionization, dissociation, and excitation (Porter et al., 1976). Also, ionization of O should play a role in the upper mesosphere and lower thermosphere, where O becomes one of the most abundant species. Is this not included in SIC? Please clarify.

**Answer 8: Authors agree with the comment and necessary corrections have been made in the text: "In the initial step, primary proton collisions with molecular nitrogen or oxygen cause dissociative ionization and produce secondary electrons which also form atomic nitrogen or oxygen from molecular nitrogen or oxygen." (page 3, lines 11-12)**

Comment 9: Methodology, pare 4, line 1: does the reduced mechanism contain all reactions of the important + necessary species, or a subset? E.g., is there an attempt to reduce/limit the number of reactions of the species included?

**Answer 9: Yes, it contains all the reactions of the important and necessary species. Clarified: "…307 ion-neutral reactions in total of all the important and necessary species" (page 3, line 5)**

Comment 10: Page 8, line 7, … neutral profiles are reproduced within a factor of 2 … in Figure 2, all neutral profiles are within the 5.

**Answer 10: corrected: "within a factor of 5" (page 8, line 19)**

Comment 11: Page 10, line 1-2: Earlier (line 6, line 21-22) you mention that the production of NOx due to ionization in the standard WACCM is parametrized to 1.25 NOx per ion pair; here you suggest that the NOx production is due to the five-ion chemistry scheme implemented in standard WACCM (for the lower thermosphere). Please clarify which is correct.

**Answer 11: This is clarified in the text now: "According to (Jackman *et al.*, 2005) an ion pair produces 1.25 N atoms with branching ratios of 0.55 N($^4$S) and 0.70 N($^2$D). Then the nitrogen atoms will go through a complex ion chemistry that leads to the formation of NOx species." (page 6, lines 28-32)**

Comment 12: Conclusions, page 12, line 13: You could include an additional statement like: Before and after the solar proton event, NO and NO2 from WACCM-rSIC agree much better with results from WACCM-SIC than the results of WACCM-D, because…

[revised manuscript text omitted]

$$NO \rightarrow NO^+ + e^-$$

$$O_2(^1D_g) \rightarrow O_2^+ + e^-$$

$$CO_2 \rightarrow CO_2^+ + e^-$$

5    $$O_3 \rightarrow O_3^+ + e^-$$

[Figure]

Figure 1. Root-mean-square (rms) error of all species as a function of number of species in the reduced mechanism for the four selected altitudes (60, 70, 80 and 90 km).

[Figure]

**Figure 2. Atmospheric concentration profiles and relative differences calculated using the full SIC model and reduced (rSIC) model for 17$^{th}$ January 2005 at 17:50 UT. In the left-hand panels the solid lines show the concentrations calculated by the full SIC model, while the symbols refer to the reduced rSIC model. The right-hand panels show the percentage difference between the rSIC and SIC models: (a) concentrations and (b) percentage differences of $HNO_3$, $NO$, $NO_2$, $O_3$ and $H_2O_2$; (c) concentrations and (d) percentage differences of $OH$ and $HO_2$; (e) concentrations and (f) percentage differences of $NO^+$ and $O_2^+$; (g) concentrations and (h) percentage differences of $CO_3^-$, $NO_3^-$ and $O_2^-$.**

(a)

[Figure]

**(b)**

[Figure]

[Figure]

[Figure]

**Figure 3. (a) Zonal mean vertical concentration profiles of mesospheric O$_3$ (in ppm) for northern hemisphere polar latitudes (60°-90°N) during January 2005; (b) percentage difference between three models (WACCM-D, WACCM and WACCM-rSIC) and the full WACCM-SIC.**

[Figure]

**(b)**

[Figure]

[Figure]

[Figure]

**Figure 4. (a) Zonal mean vertical concentration profiles of mesospheric/upper stratospheric NO (in ppb) for northern hemisphere polar latitudes (60°-90°N) during January 2005; (b) percentage difference between three models (WACCM-D, WACCM and WACCM-rSIC) and the full WACCM-SIC.**

[Figure]

**(b)**

[Figure]

[Figure]

[Figure]

**Figure 5. (a) Zonal mean vertical concentration profiles of mesospheric/upper stratospheric NO (in pp) for northern hemisphere polar latitudes (60°-90°N) during January 2005; (b) percentage difference between three models (WACCM-D, WACCM and WACCM-rSIC) and the full WACCM-SIC.**

[Figure]

**(b)**

[Figure]

[Figure]

[Figure]

**Figure 6. (a) Zonal mean vertical concentration profiles of mesospheric/upper stratospheric OH (in ppb) for northern hemisphere polar latitudes (60°-90°N) during January 2005; (b) percentage difference between three models (WACCM-D, WACCM and WACCM-rSIC) and the full WACCM-SIC.**

**(a)**

[Figure]

**(b)**

[Figure]

[Figure]

[Figure]

**Figure 7. (a) Zonal mean vertical concentration profiles of mesospheric/upper stratospheric HO$_2$ (in ppb) for northern hemisphere polar latitudes (60°-90°N) during January 2005; (b) percentage difference between three models (WACCM-D, WACCM and WACCM-rSIC) and the full WACCM-SIC.**

(a)

[Figure]

**(b)**

[Figure]

[Figure]

[Figure]

**Figure 8. (a) Zonal mean vertical concentration profiles of mesospheric/upper stratospheric HNO₃ (in ppb) for northern hemisphere polar latitudes (60°-90°N) during January 2005; (b) percentage difference between three models (WACCM-D, WACCM and WACCM-rSIC) and the full WACCM-SIC.**